# FAPEX: Fractional Amplitude-Phase Expressor for Robust Cross-Subject Seizure Prediction

**Ruizhe Zheng[1]§**
Research Institute of Intelligent Complex Systems, Fudan University
rzzheng23@m.fudan.edu.cn

**Lingyan Mao[2]§**
Department of Neurology, Zhongshan Hospital, Fudan University
lingyanmao@fudan.edu.cn

**Tian Luo[4]**
Children's Hospital of Fudan University
tianluo@fudan.edu.cn

**Yi Wang[4]**
Children's Hospital of Fudan University
yiwang@shmu.edu.cn

**Dingding Han[3] ***
School of Information Science and Technology, Fudan University
ddhan@fudan.edu.cn

**Jing Ding[2]***
Zhongshan Hospital, Fudan University
jingding@zs-hospital.sh.cn

**Yuguo Yu[1]***
State Key Laboratory of Brain Function and Disorders and MOE Frontiers Center for Brain Science,
Research Institute of Intelligent Complex Systems and Institutes of Brain Science, Fudan University,
Shanghai Artificial Intelligence Laboratory Shanghai 200232, China
yuyuguo@fudan.edu.cn

## Abstract

Precise, generalizable subject-agnostic seizure prediction (SASP) remains a fundamental challenge due to the intrinsic complexity and significant spectral variability of electrophysiologial signals across individuals and recording modalities. We propose FAPEX, a novel architecture that introduces a learnable *fractional neural frame operator* (FrNFO) for adaptive time–frequency decomposition. Unlike conventional models that exhibit spectral bias toward low frequencies, our FrNFO employs fractional-order convolutions to capture both high and low-frequency dynamics, achieving approximately $10\%$ improvement in F1-score and sensitivity over state-of-the-art baselines. The FrNFO enables the extraction of *instantaneous phase and amplitude representations* that are particularly informative for preictal biomarker discovery and enhance out-of-distribution generalization. FAPEX further integrates structural state-space modeling and channelwise attention, allowing it to handle heterogeneous electrode montages. Evaluated across 12 benchmarks spanning species (human, rat, dog, macaque) and modalities (Scalp-EEG, SEEG,

---

*Corresponding authors.

39th Conference on Neural Information Processing Systems (NeurIPS 2025).

ECoG, LFP), `FAPEX` consistently outperforms 23 supervised and 10 self-supervised baselines under nested cross-validation, with gains of up to $15\%$ in sensitivity on complex cross-domain scenarios. It further demonstrates superior performance in several external validation cohorts. To our knowledge, these establish `FAPEX` as the first epilepsy model to show consistent superiority in SASP, offering a promising solution for discovering epileptic biomarker evidence supporting the existence of a distinct and identifiable preictal state for and clinical translation.

# 1    Introduction

Epilepsy is a common, heterogeneous set of neurological disorders characterized by recurrent, hypersynchronous discharges that disrupt normal cognition and behavior. Affecting over 50 million people worldwide [57], its diagnosis and monitoring rely fundamentally on electrophysiological recordings—whether invasive (e.g., electrocorticography (ECoG), stereo-electroencephalography (SEEG), local field potential (LFP)) or non-invasive (scalp EEG) [38, 17]. Although seizures have long been viewed as abrupt and unpredictable events, a growing body of work demonstrates the existence of a preictal stage marked by subtle neural and behavioral changes, offering an actionable window for intervention.

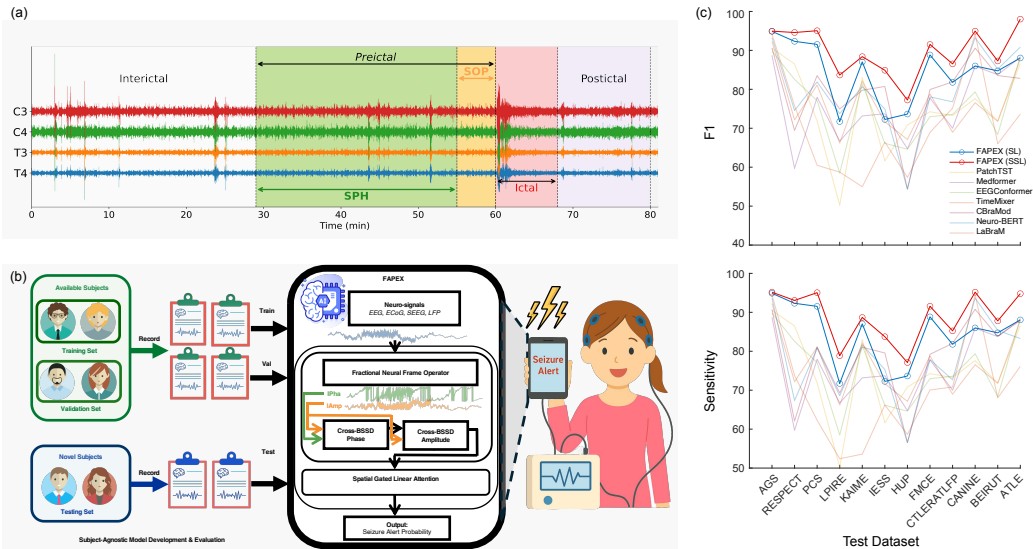

**Figure 1:** **Summary of our work.** (a) Definition of the different brain activity stages for the predictive analysis of epileptic seizure (a scalp-EEG record is shown for the purpose of illustration). (b) Overview of the `FAPEX` development and validation pipeline. (c) Comparative evaluation results demonstrate that `FAPEX` consistently outperforms state-of-the-art (SOTA) supervised and self-supervised approaches across 12 diverse benchmarks in terms of F1 and sensitivity, demonstrating superior performance and generalization.

Seizure prediction systems seek to detect these preictal alterations and raise alarms sufficiently in advance. As shown in Fig. 1 (a), within the established framework of ictogenesis - which delineates interictal, ictal, and postictal phases - the preictal interval offers a crucial target for clinical interventions ranging from simple alerts aimed at mitigating injury risk to sophisticated closed-loop neuromodulation devices. To formalize practical deployment, seizure prediction systems are typically evaluated with respect to two time parameters: the Seizure Prediction Horizon (SPH), which defines the minimum interval between a raised alarm and seizure onset to allow meaningful intervention, and the Seizure Occurrence Period (SOP), a predefined window during which a seizure is expected following an alarm.

**Why subject-agnostic seizure prediction (SASP)?** Despite remarkable advances in seizure prediction achieved by pioneering studies [61, 39, 73, 9, 15, 74, 49], the field remains constrained by two fundamental limitations: the reliance on subject-specific modeling paradigms and limited scalability. Subject-specific approaches, while often achieving impressive performance on individual patients, require extensive labeled data collection for each new patient and cannot leverage knowledge across diverse patient populations. This impedes large-scale clinical adoption and negates the potential advantages of aggregating data to identify generalizable seizure biomarkers.

Beyond subject specificity, additional obstacles include narrow EEG modality ranges [69, 39, 72], inconsistent preprocessing pipelines [8, 23], and dependence on rigid electrode configurations. Together, these factors highlight the urgent need for truly subject-agnostic predictive algorithms capable of operating robustly across various patient and recording configurations. Specific challenges include:

**(1) Capturing refined high- and low-frequency biomarkers.** Clinical evidence [46, 14, 27, 50, 47] shows pathological high-frequency oscillations and low-frequency fluctuations serve as crucial epileptogenesis biomarkers. These subtle, non-stationary features are easily obscured by artifacts. Conventional CNNs [4] and Transformers [36, 53] exhibit spectral biases toward low frequencies, struggling to preserve transient HFO signatures [63, 40].

**(2) Modeling phase–amplitude interactions.** Epilepsy exhibits abnormal phase-amplitude coupling and (de)synchronization [1, 35, 26, 2, 76, 64] during seizure initiation and propagation across frequency bands. These interactions provide critical clues for distinguishing ictal from interictal states. Current clinical models typically utilize amplitude information in time or frequency domains separately, rarely integrating both. Our architecture captures these fundamental aspects of neural oscillations, leveraging their complementary insights into seizure evolution.

**(3) Handling heterogeneous channel layouts.** Seizure onset zones and preictal activity distribution vary significantly across individuals, with predictive features appearing on different electrodes from patient to patient. Implantation strategies, montage configurations, and regional coverage introduce further variability in channel characteristics. Naively pooling signals across channels obscures patient-specific biomarkers and amplifies noise, compromising generalization in subject-agnostic contexts.

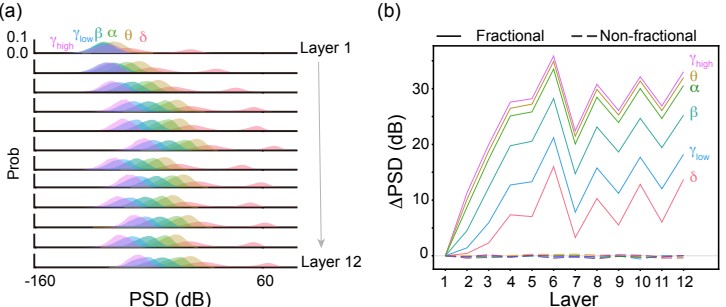

**Figure 2: Interpretability of** `FAPEX`**.** (a) Kernel density estimates of power spectral density (PSD) responses for FrNFO filters across layers and brain frequency subbands. As depth increases, the operator progressively refines its discrimination among subbands, maintaining the natural low-frequency, high-energy and high-frequency, low-energy distribution, with energy gradually stabilizing after intermediate layers. (b) Layer-wise frequency-specific gain relative to the initial layer. Unlike non-fractional operators, FrNFO consistently amplifies both low- and high-frequency components, achieving balanced cross-frequency representations, indicating its ability to capture both fast and slow neural dynamics essential for seizure prediction.

**Present work.** To overcome these challenges, we propose `FAPEX`, a unified model for effective generalization across heterogeneous EEG settings, electrode configurations, and clinical subtypes. Our approach integrates three key innovations: (1) *fractional neural frame operator* (FrNFO): a learnable bank of Weyl-Heisenberg filters for adaptive time-frequency decomposition. FrNFO extracts high-fidelity features through fractional-order convolutions with minimal spectral leakage, capturing both high and low-frequency components of epileptic signals. (2) *amplitude-phase cross-encoding* (APCE): A bidirectional state-space architecture processing phase and amplitude representations, learning time-varying relationships to extract seizure evolution patterns. (3) *Spatial correlation aggregation* (SCA): Channel-wise attention mechanisms modeling inter-electrode dependencies to identify predictive spatial patterns. Together, these components enable FAPEX to learn multi-scale representations that capture phase-amplitude coupling while handling non-stationarity and channel heterogeneity. **Together**, these components enable FAPEX to learn rich, multi-scale representations that capture subtle changes in phase-amplitude coupling across frequencies while adaptively handling non-stationarity and channel heterogeneity. As illustrated in Fig. 2, FrNFO serves as the foundational component, addressing low-frequency bias by preserving fragile yet critical high-frequency oscillations while providing fine-grained decomposition of amplitude and phase features for a more comprehensive picture of neural activity in seizure prediction.

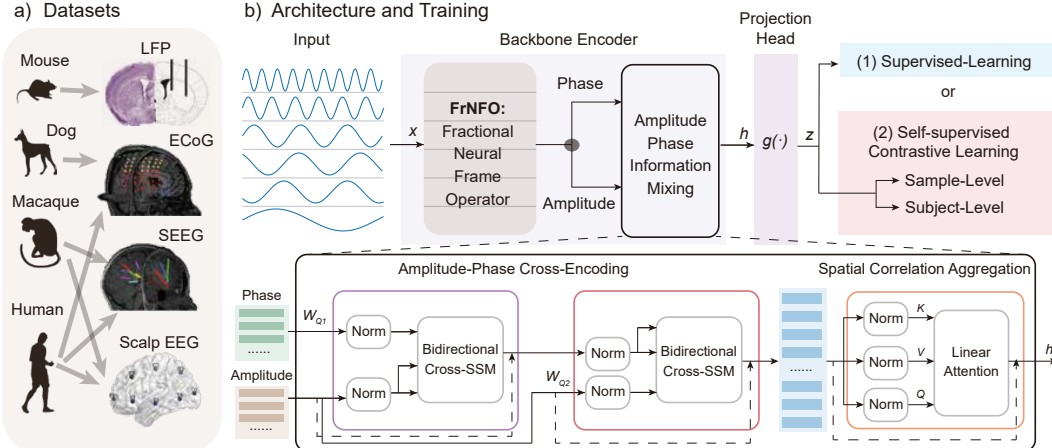

**Figure 3:** Datasets and network architecture summarization. (a) We used LFP, ECoG, SEEG and Scalp EEG data across species (humans, dogs, rats, and macaques), to validate our model. (b) The network structure and training pipeline of our FAPEX framework. The input signals will be encoded by the backbone encoder that is consisted of our FrNFO, naturally separated into phase and amplitude sections, then go through a Amplitude-Phase information mixing procedure which deals with the two sections interactively using 2 biderectional Cross-SSM modules, and use a linear attention module for spatial correlation aggregation.

Our main innovations are: **(1)** FAPEX, a subject-agnostic framework integrating our novel Fractional Neural Frame Operator (FrNFO), amplitude-phase cross-encoding, and spatial correlation aggregation to anticipate seizures across diverse modalities. **(2)** FrNFO, a learnable bank of Weyl–Heisenberg filters that performs fractional-order time-frequency decomposition, mitigates low-frequency bias and preserves high-frequency oscillations with provable robustness. **(3)** Extensive validation on 12 benchmarks across species and recording modalities shows FAPEX consistently outperforms 32 baselines, establishing a new standard for seizure prediction and revealing meaningful preictal biomarkers.

## 2 Method

**Problem formulation.** Epileptologists classify seizure dynamics into three phases: interictal, preictal, and ictal. **Interictal Phase**: Periods between seizures (typically $> 30$ minutes) with generally normal brain activity, though occasional interictal epileptiform discharges may occur. **Preictal Phase**: The period preceding a seizure, marked by subtle brain activity changes that may predict an impending seizure. **Ictal Phase**: The seizure event itself, characterized by ictal epileptiform discharges. Formally, a neuroelectrical segment is a set of time series $\{\boldsymbol{x}^{(i)}\}_{i=1}^{C}$, where $C$ is the number of channels, and each $\boldsymbol{x}^{(i)} \in \mathbb{R}^{T}$ represents a channel with $T$ timestamps. A seizure predictor constructs a function $f_{\text{model}}$ that maps $\{\boldsymbol{x}^{(i)}\}_{i=1}^{C}$ to a binary label $\hat{y}_i$, distinguishing interictal from preictal states. The model is trained to align predictions with clinical annotations $y_i$, enabling prediction in unseen subjects.

**Input patchifying.** Given a multichannel neural signal segment $\boldsymbol{X} \in \mathbb{R}^{C \times T}$, where $C$ represents the number of electrode channels and $T$ is the total number of time samples, we aim to establish a preprocessing pipeline that is robust to variations in electrode count and placement. To this end, the continuous data is first partitioned into fixed-duration, non-overlapping patches. Specifically, each channel signal $\mathbf{x}_c \in \mathbb{R}^{T}$ is segmented using a window of length $\tau$, resulting in $N = \lfloor \frac{T}{\tau} \rfloor$ patches per channel. Each patch is then projected into a common feature space using a channel-shared linear embedding $\boldsymbol{W} \in \mathbb{R}^{d_{\text{model}} \times \tau}$ and bias vector $\mathbf{b} \in \mathbb{R}^{d_{\text{model}}}$, resulting in the embedded tensor $\tilde{\boldsymbol{X}} \in \mathbb{R}^{C \times N \times d_{\text{model}}}$ ($d_{\text{model}}$ is dimension of the model). This process ensures that the subsequent layers can operate independently of electrode count and spatial arrangement.

### 2.1 Fractional neural frame operator (FrNFO)

**Motivation.** Nonstationary signals, such as those encountered in neuroelectrical recordings in epilepsy patients, present significant challenges due to their highly variable time-frequency content

and variability in both amplitude and phase. The *Fractional Fourier Transform (FrFT)* has emerged as a powerful tool for analyzing such signals, providing a flexible, continuous interpolation between the time domain ($\theta = 0$) and the frequency domain ($\theta = \frac{\pi}{2}$) via a fractional order parameter $\theta$. Formally, FrFT generalizes the Fourier transform with a fractional order $\theta \in (0, \pi)$, defined for a signal $f \in L^2(\mathbb{R})$ as:

$$\mathcal{F}_\theta(f)(x) = \frac{1}{\sqrt{|\sin\theta|}} \int_{\mathbb{R}} f(t) \exp\left[\pi i \left((t^2 + x^2)\cot\theta - 2xt\csc\theta\right)\right] \mathrm{d}t. \tag{1}$$

This transform supports operators like $\theta$-shift, $T_s^\theta f(t) = \exp\left(2\pi i s(t - s)\cot\theta\right) f(t - s)$, and $\theta$-modulation, $M_s^\theta f(t) = \exp\left(\pi i \left(s^2\cot\theta + 2st\csc\theta\right)\right) f(t)$, enabling $\theta$-fractional convolution [3, 71]:

$$(f \star_\theta g)(x) = \frac{1}{\sqrt{|\sin\theta|}} \int_{\mathbb{R}} f(s)(T_s^\theta g)(x)\mathrm{d}s, \tag{2}$$

which offers an alternative to traditional Fourier transform-based convolutions. While the fractional Fourier transform (FrFT) provides a flexible framework for interpolating between time and frequency domains, its practical implementations face two critical challenges that limit their effectiveness for neuroelectrical signals: **(1) Chirp response constraint**: traditional FrFT relies on a fixed chirp function, imposing a globally isotropic structure that poorly adapts to the diverse, localized frequency characteristics of real-world data [22]. This restricts FrFT's expressiveness essential for nuanced phase-amplitude representation. While recent methods have introduced trainable fractional orders [68, 25], they inherit this fundamental limitation, lacking the flexibility to accommodate rapid spectral transitions and localized nonstationarities. **(2) Deformation Sensitivity**: Despite its adaptability in fractional order, FrFT remains sensitive to small deformations, including time shifts, scaling variations, and localized perturbations, which are especially prevalent in neural signals [42, 29]. These limitations underscore the need for more expressive, adaptive frameworks that can capture the intricate amplitude-phase representation.

**A neural approach for fine-grained amplitude-phase representation.** To overcome these limitations, we propose the *fractional neural frame operator*, which integrates neural implicit representations to learn a parameterization of $\theta$-fractional version of nonstationary Weyl-Heisenberg frame [20, 45, 18], defined as:

$$\Psi_\theta = \left\{ M_{lp_0^{(j)}}^\theta T_{sq_0^{(j)}}^\theta I\Phi_j : s \in \mathbb{R}, l \in \mathbb{Z}, j \in \{1, \ldots, N\} \right\}, \tag{3}$$

where $p_0^j, q_0^j$ are positive constants adjusting the scale. It involves $\theta$-modulation $M_{lp_0^{(j)}}^\theta(t) = e^{\pi i((lp_0^{(j)})^2 \cot\theta + 2lp_0^{(j)} t \csc\theta)}$ and $\theta$-shift $T_{sq_0^{(j)}}^\theta(t) = e^{2\pi i sq_0^{(j)}(t - sq_0^{(j)})\cot\theta} \Phi_j(t - sq_0^{(j)})$. $\Psi_\theta$ presents a redundant set of basis functions that can be used to represent or analyze a signal on the fractional domain. Unlike FrFT, it is equipped with adaptive windows $\Psi_j$ over each scale $j$ to capture a wide range of signal behaviors. Building upon this, we propose fractional neural frame operator.

The core of the FrNFO is an implicit multilayer perceptron (MLP) [34, 62] designed to generate adaptive window function for the frame filters. Given temporal samples $N$ and feature channels $d_{\text{model}}$, the implicit MLP defines the window kernel $\boldsymbol{\Phi} \in \mathbb{C}^{N \times d_{\text{model}}}$ for $j = 1, \ldots, N,\ k = 1, \ldots, d_{\text{model}}$ as

$$\boldsymbol{\Phi}^{j,k}(t_j) = \left( \sum_{i=1}^{M} w_{i,k} \exp(-j(b_{i,k}t_j + c_{i,k})) \right) \cdot \left( \sum_{n=0}^{K} a_{n,k} H_n(t_j) \right), \tag{4}$$

where $w_{i,k}, b_{i,k}, c_{i,k}, a_{n,k}$ are trainable parameters optimized through gradient descent. The basis functions $H_n(t) = (-1)^n e^{t^2} \frac{d^n}{dt^n} e^{-t^2}$ are Hermite polynomials, embedding prior knowledge of localized oscillatory behavior, while the sine activation functions promote smooth and periodic kernel characteristics essential for identifying quasiperiodic activities in brain.

FrNFO further introduces a learnable fractional order $\boldsymbol{\theta} = [\theta_1, \ldots, \theta_{d_{\text{model}}}] \in (0, \pi)^{d_{\text{model}}}$, which governs the time-frequency representation for each feature channel independently. Given an input neural embedding $\boldsymbol{X} \in \mathbb{C}^{N \times d_{\text{model}}}$, employing the fractional convolution theorem [3, 71], the output feature for channel $k$ is defined as:

$$\hat{\boldsymbol{X}}_{:,k} = \exp(-\pi i \omega^2 \cot\theta_k) \odot \mathcal{F}_{\theta_k}(\boldsymbol{X}_{:,k}) \odot \mathcal{F}_{\theta_k}(\boldsymbol{\Psi}_{:,k}), \quad k = 1, \ldots, d_{\text{model}}, \tag{5}$$

where $\boldsymbol{\Psi}_{:,k}$ is the frame filter kernel equipped with learnable window kernel, $\odot$ denotes the Hadamard product, and $\omega$ represents the frequency grid. The phase adjustment factor $\exp(\pi i \omega^2 \cot \theta_k)$ ensures proper alignment and interpretation of fractional frequency components. This adaptive formulation allows FrNFO to dynamically adjust frequency resolution.

**FrNFO is a provably robust amplitude representator.** As previously formulated, as a neural fractional-order filterbank, FrNFO naturally yields complex-valued signal representation that can be easily formulated into phases and amplitudes across different scales and fractional orders. We further highlight that it also provides a provably robust amplitude representation, which is the main information source in many applications, from the perspective of scattering transform. Refer to further discussion and proof in App. A.

## 2.2 Amplitude-phase encoding

**Amplitude-phase cross encoding (APCE).** We introduce APCE to capture heterogeneous, cross-frequency dependencies between amplitude and phase embeddings produced by FrNFO. Inspired by recent advances in selective state space model, proposed first in Mamba [**? ? ? ?** ], we adopt a bidirectional state-space mechanism building on Mamba blocks with cross-attention-like mechanism [59], as shown in Fig. 3. Formally, given amplitude embeddings $\mathbf{Amp}$ and phase embeddings $\mathbf{Pha}$, we normalize them as: $\mathbf{Amp} = \mathrm{RMSNorm}(\mathbf{Amp}), \quad \mathbf{Pha} = \mathrm{RMSNorm}(\mathbf{Pha})$. These normalized embeddings are then processed by the dual cross-Mamba module, which operates in a channel-independent manner to capture amplitude-phase interactions using a bidirectional state-space model (BSSM), comprising two sequential blocks: *phase BSSM* and *amplitude BSSM*. In the phase BSSM block, the normalized phase embeddings $\mathbf{Pha} \in \mathbb{R}^{B \times M \times D}$ are projected into a latent space via two shared linear mappings:

$$\boldsymbol{X}^P = \boldsymbol{W}^x \mathbf{Pha}, \quad \boldsymbol{Z}^P = \boldsymbol{W}^z \mathbf{Pha}, \tag{6}$$

where $\boldsymbol{W}^x, \boldsymbol{W}^z \in \mathbb{R}^{D \times E}$ are learnable projection matrices, and $E$ denotes the number of latent SSM states. The projected embeddings undergo causal and anti-causal convolutions followed by a SiLU activation:

$$\boldsymbol{X}_o = \mathrm{SiLU}\big(\mathrm{Conv1d}_o(\boldsymbol{X}^P)\big), \quad o \in \{\text{forward}, \text{backward}\}. \tag{7}$$

Using the normalized amplitude embeddings $\mathbf{Amp}$, we compute state-space parameters:

$$\boldsymbol{B}_o = \boldsymbol{W}^B \mathbf{Amp}, \quad \boldsymbol{C}_o = \boldsymbol{W}^C \mathbf{Amp}, \quad \boldsymbol{\Delta}_o = \mathrm{Softplus}\big(\boldsymbol{W}^\Delta \boldsymbol{X}_o + \boldsymbol{b}^\Delta\big), \tag{8}$$

where $\boldsymbol{W}^B, \boldsymbol{W}^C \in \mathbb{R}^{D \times N}$ are shared across directions, and $\boldsymbol{W}^\Delta \in \mathbb{R}^{E \times E}, \boldsymbol{b}^\Delta \in \mathbb{R}^E$ are shared scaling parameters. The time-varying transition parameters are then defined as:

$$\overline{\boldsymbol{A}}_o = \boldsymbol{\Delta}_o \otimes \boldsymbol{A}, \quad \overline{\boldsymbol{B}}_o = \boldsymbol{\Delta}_o \otimes \boldsymbol{B}_o, \tag{9}$$

where $\boldsymbol{A} \in \mathbb{R}^{E \times N}$ is a shared, direction-agnostic transition matrix, and $\otimes$ denotes element-wise multiplication. The output sequence is computed using the SSM kernel:

$$\boldsymbol{Y}_o = \mathrm{SSM}(\overline{\boldsymbol{A}}_o, \overline{\boldsymbol{B}}_o, \boldsymbol{C}_o)(\boldsymbol{X}_o). \tag{10}$$

The final phase-to-amplitude representation, which captures phase-informative patterns, is gated as:

$$\boldsymbol{Y}^P = (\boldsymbol{Y}_{\text{forward}} + \boldsymbol{Y}_{\text{backward}}) \odot \mathrm{SiLU}(\boldsymbol{Z}^P). \tag{11}$$

In the amplitude BSSM, the roles are swapped: the phase-informative $\boldsymbol{Y}^P$ provides the context, and the amplitude $\mathbf{Amp}$ serves as queries. A residual connection combines the block output $\boldsymbol{Y}^A$ with the original amplitude embeddings $\mathbf{Amp}$ to produce the final APCE encoding:

$$\tilde{\boldsymbol{X}} = \boldsymbol{Y}^A + \mathbf{Amp}, \quad \tilde{\boldsymbol{X}} \in \mathbb{R}^{B \times M \times D}. \tag{12}$$

**Spatial correlation aggregation (SCA).** During the preictal interval, epilepsy is marked by dynamic shifts in inter-electrode interdependencies that reflect the spread of pathological activity across brain regions. Accurate seizure forecasting from multichannel recordings therefore hinges on modeling these spatial dependencies. To this end, given neuroelectrical embeddings $\boldsymbol{X} \in \mathbb{R}^{C \times N \times d}$, SCA models global cross-spatial dependencies of different electrodes while integrating local spatiotemporal

patterns. Formally, linear attention aims to use $\phi\left(\mathbf{q}_i\right)\phi\left(\mathbf{k}_j\right)^\top$ to approximate softmax attention kernel at linear complexity, where the feature map $\phi(\cdot):\mathbb{R}^d\mapsto\mathbb{R}^d$ is applied row-wise to the query and key matrices. As a result, the $c$-th row of attention output $\mathbf{a}_t\in\mathbb{R}^d$ can be rewritten as

$$\mathbf{o}_c=\mathbf{a}_c\odot\text{Sigmoid}(\mathbf{g}_c),\quad \mathbf{a}_c=\frac{\sum_{i=1}^C\phi\left(\mathbf{q}_c\right)\phi\left(\mathbf{k}_i\right)^\top\mathbf{v}_i}{\sum_{j=1}^C\phi\left(\mathbf{q}_c\right)\phi\left(\mathbf{k}_j\right)^\top}=\frac{\phi\left(\mathbf{q}_c\right)\sum_{i=1}^C\phi\left(\mathbf{k}_i\right)^\top\mathbf{v}_i}{\phi\left(\mathbf{q}_c\right)\sum_{j=1}^C\phi\left(\mathbf{k}_j\right)^\top},\quad(13)$$

where $\mathbf{g}_c$ is the $c$-th row of $\boldsymbol{G}:=\text{RMSNorm}[\text{DepthwiseConv2d}\left(\boldsymbol{X}\right)]$ implemented with a $3\times3$ depthwise convolutional kernel to aggregate neighborhood spatiotemporal information with RM-SNorm to improve stability. The feature map $\phi$ is made as a one-layer MLP as $\phi_{\text{MLP}}(\boldsymbol{x}):=\exp(\boldsymbol{W}_1^\top\boldsymbol{x})$, where the matrix $\boldsymbol{W}_1,\boldsymbol{W}_2\in\mathbb{R}^{d\times d}$.

## 3 Experiments

We conducted empirical investigations to address the following **Research Questions**: **RQ1**: How does FAPEX perform in SASP relative to supervised baselines? **RQ2**: Does self-supervised pretraining improve performance of FAPEX in SASP relative to self-supervised baselines? **RQ3**: How well does FAPEX generalize to different cohorts (*e.g.*, species, institution)? **RQ4**: What is the contribution of each design choice within FAPEX?

### 3.1 Experimental settings

We evaluate FAPEX across diverse settings spanning supervised learning (**RQ1**), self-supervised pretraining-finetuning (**RQ2**), and cross-cohort transfer (**RQ3**). This section outlines the baseline, evaluation protocols, and other basic implementation setups common to all experiments. See details of training protocols in App. G. Full implementation details are provided in App. H.

**Datasets.** We compile 12 benchmarking datasets spanning four species (human, rat, dog, macaque) and multiple acquisition modalities (Scalp-EEG, ECoG, SEEG, LFP) for evaluation, as summarized in Tab. 1. All recordings are resampled and segmented to standardized lengths, then harmonized to 64 effective channels via channel rejection and duplication, enabling consistent input formatting across all models. See detailed descriptions and preprocessing procedures in App. F. Note that we apply channel alignment during preprocessing to facilitate consistent training across diverse datasets for both our model and a broad range of baselines. In short, FAPEX itself is inherently agnostic to the number and configuration of input channels.

**Table 1: Summary of datasets.** The datasets span several species (human, rat, dog, macaque) and acquisition modalities (Scalp-EEG, ECoG, SEEG, LFP).

| Dataset | Confidentiality | Species | # Subj. | Modality | # Ch. | # Samples | Duration | SOP | SPH | ID/IV | OOD/EV |
|---|---|---|---|---|---|---|---|---|---|---|---|
| FMCE | Public | Human | 65 | ECoG/SEEG[1] | 64 | 32,323 | 4 s | 30 s | 1 min | ✓ | ✗ |
| HUP | Public | Human | 73 | ECoG/SEEG | 64 | 53,323 | 4 s | 30 s | 5 min | ✓ | ✗ |
| RESPECT | Public | Human | 6 | ECoG | 64 | 17,214 | 4 s | 30 s | 5 min | ✓ | ✓ |
| BEIRUT | Public | Human | 6 | Scalp-EEG | 64 | 35,941 | 4 s | 1 min | 30 min | ✓ | ✓ |
| CTLE-RATLFP | Public | Rat | 7 | LFP | 64 | 11,732 | 2 s | 30 s | 5 min | ✓ | ✗ |
| LPIRE | Public | Rat | 15 | LFP | 64 | 159,715 | 2 s | 30 s | 5 min | ✓ | ✓ |
| CANINE | Public | Dog | 6 | ECoG | 64 | 382,278 | 4 s | 5 min | 4 hr | ✓ | ✓ |
| ATLE | Private | Human | 5 | Scalp-EEG | 64 | 11,536 | 4 s | 5 min | 30 min | ✓ | ✗ |
| AGS | Private | Human | 5 | Scalp-EEG | 64 | 32,323 | 4 s | 5 min | 30 min | ✓ | ✓ |
| IESS | Private | Human | 17 | Scalp-EEG | 64 | 48,986 | 4 s | 5 min | 30 min | ✓ | ✓ |
| KAIME | Private | Macaque | 3 | Scalp-EEG & SEEG[2] | 64 | 36,092 | 4 s | 5 min | 30 min | ✓ | ✓ |
| PCS | Private | Human | 5 | Scalp-EEG | 64 | 29,679 | 4 s | 5 min | 30 min | ✓ | ✗ |
| TUEG | Public | Human | 14,987 | Scalp-EEG | 64 | 1,030,090 | 32 s | *Used for Pretraining Only* | | | |
| CCEP | Public | Human | 74 | ECoG | 64 | 52,337 | 32 s | *Used for Pretraining Only* | | | |
| PPE | Public | Human | 30 | Scalp-EEG | 64 | 13,434 | 32 s | *Used for Pretraining Only* | | | |

[1] For "ECoG/SEEG" datasets each subject has *either* sub-dural ECoG grids/strips *or* SEEG depth electrodes, never both.
[2] KAIME comprises simultaneous scalp-EEG and SEEG depth recordings from three adult rhesus macaques (*Macaca mulatta*).

**Baselines.** We compare our method against the following baselines, including 22 supervised baselines for **RQ1** and 10 self-supervised ones for **RQ2**. The supervised baselines include (1) *Convolutional models* (5 baselines): ModernTCN [10], MRConv [7], MultiresNet [43], Omni-Scale [48], and SPaRCNet [21]; (2) *Token mixers* (6 baselines): EEGConformer [44], iTransformer [32], Nonformer [31], PatchTST [37], Pathformer [6], and SeizureFormer [13]; (3) *Time-frequency mixers* (4 baselines): ATFNet [66], FreTS [67], NFM [24], and TSLANet [12]; (4) *Multiscale token mixers* (7 baselines): AdaWaveNet [?], Medformer [55], MTST [75], Pyraformer [30], SimpleTM [5], TimesNet [60], and TimeMixer [54]. Self-supervised baselines include 6 *non-contrastive* models:

Brant [72], CBraMod [52], EEGPT [51], LaBraM [19], Neuro-BERT [58], VQ_MTM [16]; 4 *contrastive* models: BIOT [65], COMETS [56], MF-CLR [11], and TS2Vec [70]. See details in App. E.

**Evaluation protocols.** All experiments follow a consistent *subject-agnostic nested cross-validation* (SANCV) scheme. For each dataset, subjects are split into non-overlapping train, validation, and test folds. These folds are randomly permuted to yield multiple experimental runs for **RQ1-3**. For **RQ1**, we evaluate in-domain performance with full supervision. For **RQ2**, we evaluate in-domain performance by supervised finetuning. for **RQ3**, we evaluate out-of-domain performance on several regimes for our approach and two best-performing self-supervised baselines. We report median and interquartile range (IQR) across runs for: Balanced Accuracy (BA), Sensitivity (SEN), F1, AUROC, AUPRC. We also report Stratified Brier Score to indicate both discriminative and calibration quality.We calculate F1 as the monitoring score as it captures the trade-off between reducing false alarms and maintaining high sensitivity. We adopt the Friedman test as a nonparametric omnibus for statistical significance with Bayesian *post hoc* comparison. Refer to App. H for details.

**Table 2: Median performance across publicly available datasets.** Top-1, Top-2, and Top-3 results are highlighted in red, blue, and green, respectively, within both supervised (SL) and self-supervised (SSL) groups. **FAPEX** demonstrates consistently strong performance, achieving top-1 TO 3 rankings on the majority of datasets and metrics, reflecting its generalization and adaptability. For detailed results and statistical analysis, refer to App. C.

| Model | BEIRUT SEN | F1 | ROC | CANINE SEN | F1 | ROC | FMCE SEN | F1 | ROC | cTLE-RatLFP SEN | F1 | ROC | LPIRE SEN | F1 | ROC | HUP SEN | F1 | ROC | RESPECT SEN | F1 | ROC |
|---|---|---|---|---|---|---|---|---|---|---|---|---|---|---|---|---|---|---|---|---|---|
| ModernTCN | 83.4 | 83.1 | 85.0 | 84.8 | 84.0 | 73.4 | 79.6 | 89.3 | 88.7 | 69.2 | 74.0 | 89.5 | 68.7 | 72.6 | 80.3 | 71.3 | 70.3 | 67.3 | 70.0 | 75.0 | 80.0 |
| MRConv | 78.6 | 78.2 | 83.5 | 83.7 | 83.8 | 72.6 | 78.8 | 89.9 | 88.2 | 73.0 | 76.4 | 73.0 | 60.5 | 68.4 | 68.7 | 69.1 | 67.8 | 66.1 | 71.1 | 72.7 | 73.8 |
| MultiresNet | 73.3 | 72.8 | 74.7 | 64.8 | 72.0 | 70.8 | 75.8 | 82.5 | 83.3 | 63.0 | 68.8 | 87.1 | 67.1 | 71.7 | 80.1 | 65.3 | 63.8 | 65.5 | 62.4 | 75.2 | 61.7 |
| Omni-Scale | 72.6 | 71.5 | 83.1 | 65.3 | 71.9 | 73.2 | 79.1 | 84.1 | 83.7 | 75.8 | 78.3 | 72.5 | 51.9 | 65.1 | 75.7 | 70.7 | 68.7 | 67.1 | 69.2 | 73.3 | 75.7 |
| SPaRCNet | 71.1 | 71.6 | 79.1 | 85.9 | 84.7 | 74.0 | 60.4 | 67.7 | 65.9 | 72.2 | 75.5 | 67.5 | 43.6 | 48.2 | 50.8 | 61.6 | 60.2 | 62.8 | 73.7 | 81.3 | 63.0 |
| EEGConformer | 68.4 | 66.5 | 82.5 | 79.3 | 81.4 | 52.9 | 73.0 | 85.0 | 81.4 | 73.5 | 76.8 | 72.3 | 58.5 | 65.8 | 70.2 | 64.6 | 63.3 | 64.6 | 82.5 | 84.6 | 86.5 |
| EEGMamba | 70.0 | 68.5 | 82.6 | 79.6 | 81.2 | 53.2 | 68.9 | 95.5 | 85.7 | 62.0 | 68.0 | 89.5 | 63.7 | 70.7 | 80.7 | 63.2 | 61.3 | 62.8 | 79.6 | 80.0 | 73.0 |
| iTransformer | 70.6 | 69.2 | 82.8 | 64.2 | 70.4 | 71.2 | 68.4 | 80.0 | 78.3 | 56.2 | 62.6 | 75.6 | 40.9 | 46.0 | 49.4 | 58.3 | 57.7 | 57.4 | 80.5 | 84.7 | 87.9 |
| Nonformer | 68.6 | 63.4 | 75.2 | 60.7 | 64.9 | 70.3 | 74.9 | 82.6 | 86.5 | 72.6 | 76.3 | 80.9 | 64.2 | 74.4 | 76.1 | 62.1 | 61.9 | 62.4 | 74.1 | 78.7 | 81.6 |
| PatchTST | 71.9 | 72.5 | 79.3 | 77.6 | 80.6 | 72.8 | 74.2 | 91.8 | 86.6 | 73.3 | 76.5 | 72.0 | 50.2 | 50.4 | 52.6 | 70.8 | 69.3 | 67.5 | 86.5 | 86.9 | 82.2 |
| Pathformer | 67.6 | 64.1 | 81.7 | 65.9 | 72.3 | 71.8 | 77.9 | 91.2 | 88.9 | 80.4 | 81.3 | 82.8 | 68.7 | 73.0 | 80.1 | 65.7 | 65.2 | 66.3 | 76.9 | 79.9 | 82.7 |
| SeizureFormer | 67.0 | 60.2 | 79.8 | 78.8 | 81.8 | 53.5 | 73.6 | 78.8 | 79.3 | 56.6 | 63.0 | 84.5 | 65.6 | 73.2 | 68.2 | 59.7 | 59.4 | 61.0 | 83.1 | 85.0 | 74.3 |
| ATFNet | 76.0 | 74.6 | 79.4 | 62.8 | 70.9 | 71.3 | 73.4 | 81.3 | 82.7 | 54.1 | 60.7 | 78.2 | 40.0 | 44.9 | 50.0 | 64.5 | 60.4 | 63.6 | 78.4 | 82.9 | 85.7 |
| FreTS | 64.7 | 58.6 | 81.8 | 46.8 | 54.3 | 69.5 | 62.0 | 68.3 | 61.4 | 49.2 | 56.5 | 68.5 | 34.9 | 38.6 | 49.4 | 62.6 | 50.1 | 57.2 | 64.0 | 72.5 | 78.4 |
| NFM | 77.3 | 75.7 | 79.7 | 71.5 | 76.9 | 72.0 | 74.4 | 91.8 | 86.4 | 46.5 | 52.5 | 74.0 | 37.3 | 43.0 | 48.8 | 62.7 | 63.4 | 64.4 | 75.0 | 77.0 | 80.8 |
| TSLANet | 83.4 | 83.1 | 85.0 | 85.7 | 84.4 | 73.0 | 74.2 | 91.8 | 86.6 | 65.3 | 70.7 | 88.9 | 68.6 | 72.2 | 79.9 | 67.9 | 66.5 | 67.5 | 74.3 | 78.7 | 76.2 |
| AdaWaveNet | 68.0 | 66.1 | 82.7 | 79.8 | 81.3 | 52.9 | 76.6 | 82.6 | 87.3 | 55.2 | 61.4 | 83.6 | 54.1 | 66.2 | 76.2 | 64.0 | 63.2 | 64.4 | 78.2 | 84.1 | 66.6 |
| Medformer | 83.8 | 83.1 | 84.4 | 85.9 | 84.5 | 74.1 | 77.8 | 83.6 | 88.6 | 70.0 | 74.3 | 86.7 | 66.7 | 72.3 | 80.6 | 64.8 | 64.3 | 65.0 | 59.7 | 68.0 | 70.5 |
| MTST | 80.3 | 78.4 | 84.1 | 68.0 | 74.5 | 72.5 | 75.5 | 89.3 | 87.7 | 70.8 | 74.7 | 80.7 | 45.3 | 49.6 | 50.4 | 65.9 | 64.4 | 66.2 | 79.4 | 82.9 | 82.2 |
| Pyraformer | 82.8 | 81.7 | 85.4 | 80.8 | 82.3 | 72.7 | 67.0 | 96.3 | 79.8 | 60.9 | 66.9 | 86.5 | 60.2 | 72.1 | 76.2 | 60.5 | 58.2 | 58.9 | 64.0 | 72.5 | 78.4 |
| SimpleTM | 82.6 | 82.0 | 83.4 | 82.8 | 83.3 | 72.5 | 74.4 | 80.5 | 80.6 | 72.5 | 76.0 | 70.9 | 47.5 | 52.0 | 51.1 | 67.8 | 65.2 | 68.5 | 74.5 | 77.7 | 70.0 |
| TimesNet | 70.3 | 70.6 | 78.4 | 67.3 | 74.1 | 72.3 | 67.0 | 74.6 | 73.4 | 63.4 | 69.0 | 74.2 | 47.1 | 49.4 | 51.5 | 65.9 | 61.7 | 65.4 | 64.2 | 63.4 | 76.9 |
| TimeMixer | 71.8 | 72.3 | 79.0 | 76.6 | 80.0 | 73.0 | 78.8 | 82.9 | 84.5 | 69.0 | 73.8 | 87.9 | 66.2 | 72.0 | 80.6 | 67.1 | 66.2 | 67.3 | 72.1 | 76.5 | 81.9 |
| **FAPEX-Small** (SL) | 83.9 | 83.8 | 85.2 | 85.8 | 84.7 | 74.2 | 87.4 | 90.6 | 97.0 | 76.9 | 80.2 | 89.9 | 69.3 | 73.4 | 81.2 | 72.6 | 72.0 | 78.3 | 86.2 | 87.1 | 89.3 |
| **FAPEX-Base** (SL) | 84.7 | 84.3 | 85.8 | 86.0 | 84.7 | 74.5 | 88.8 | 90.7 | 97.2 | 81.8 | 83.2 | 91.2 | 71.7 | 76.1 | 81.0 | 73.7 | 72.5 | 79.3 | 92.3 | 92.3 | 91.6 |
| Brant | 71.0 | 71.7 | 78.9 | 93.2 | 92.7 | 96.6 | 75.2 | 74.7 | 86.8 | 72.1 | 76.3 | 83.6 | 56.8 | 68.6 | 75.8 | 64.0 | 63.2 | 63.9 | 70.6 | 70.4 | 76.0 |
| CBraMod | 83.9 | 83.5 | 85.5 | 90.8 | 90.6 | 98.7 | 79.2 | 79.9 | 88.8 | 82.0 | 81.9 | 82.8 | 69.3 | 74.9 | 68.4 | 56.5 | 54.4 | 79.6 | 62.1 | 69.5 | 67.6 |
| EEGPT | 63.9 | 73.3 | 71.9 | 93.4 | 92.9 | 98.5 | 68.8 | 68.9 | 74.7 | 58.7 | 64.6 | 86.2 | 65.1 | 71.5 | 80.5 | 61.3 | 58.8 | 58.8 | 76.8 | 82.2 | 60.4 |
| Neuro-BERT | 85.4 | 85.2 | 86.9 | 93.7 | 93.2 | 96.8 | 77.9 | 78.1 | 87.7 | 72.7 | 76.8 | 85.5 | 68.6 | 73.8 | 81.5 | 56.5 | 54.4 | 79.6 | 67.3 | 74.6 | 82.4 |
| VQ_MTM | 74.5 | 75.0 | 82.7 | 87.9 | 85.5 | 94.6 | 73.7 | 74.3 | 82.3 | 72.6 | 75.8 | 69.7 | 50.7 | 55.5 | 63.7 | 62.1 | 63.5 | 64.9 | 66.8 | 67.3 | 81.7 |
| COMETS | 86.2 | 86.0 | 87.3 | 93.9 | 93.4 | 98.2 | 76.1 | 76.3 | 87.1 | 74.2 | 77.4 | 83.7 | 53.4 | 61.6 | 68.6 | 65.2 | 64.1 | 64.4 | 76.3 | 81.5 | 75.6 |
| MF-CLR | 78.2 | 76.5 | 83.8 | 91.2 | 90.0 | 97.0 | 79.5 | 80.1 | 88.5 | 66.6 | 71.8 | 88.5 | 47.6 | 51.3 | 50.6 | 67.1 | 66.2 | 67.3 | 76.3 | 82.2 | 76.8 |
| TS2Vec | 57.1 | 67.0 | 58.7 | 94.7 | 94.5 | 97.5 | 75.4 | 75.7 | 77.8 | 72.0 | 75.3 | 77.8 | 59.5 | 64.1 | 51.8 | 58.9 | 51.6 | 56.7 | 79.6 | 82.6 | 78.5 |
| **FAPEX-Small** | 87.5 | 86.7 | 90.0 | 94.1 | 93.7 | 98.4 | 89.5 | 89.6 | 97.5 | 84.8 | 86.2 | 90.4 | 72.0 | 75.8 | 81.2 | 72.0 | 72.1 | 79.4 | 79.0 | 82.3 | 89.8 |
| **FAPEX-Base** | 87.8 | 87.3 | 89.9 | 95.2 | 94.9 | 99.7 | 91.5 | 91.5 | 98.0 | 85.2 | 86.5 | 90.6 | 78.9 | 83.7 | 75.2 | 77.1 | 77.2 | 81.1 | 93.1 | 94.6 | 95.2 |

## 3.2 Main results

**Performance comparison.(RQ1 and RQ2).** Tab. 2 and 3 present the results for supervised and self-supervised pretraining regimes. Across 12 datasets, our approach achieves top-1 Sensitivity (SEN) and F1 scores on all 12 datasets and top-1 ROC on 10 out of 12 datasets under the subject-dependent setup. These results demonstrate its robust capability in predicting seizure events across diverse scenarios, encompassing variations in electrophysiological recording techniques, seizure cohort etiologies, and even species. Notably, FAPEX benefits significantly from pretraining on large-scale unannotated data. It surpasses state-of-the-art foundation models, including CBraMod, VQ_MTM, and Neuro-BERT, when pretrained on the same data corpus, indicating that its performance gains stem from the model architecture rather than solely from unsupervised pretraining.

**Transferability and generalization Analysis (RQ3).** Out-of-domain validation is critical for reliable seizure prediction, requiring models to generalize across species, recording conditions, and acquisition protocols. Despite the advantages of self-supervised pretraining, generalizing to unseen domains for seizure prediction remains underexplored. We assess model transferability across diverse source-target dataset pairs to capture realistic inter-domain variability with progressively stronger supervision and adaptation: (1) Source-only transfer (SOT); (2) DIVERSIFY [33], an unsupervised domain generalization method specifically tailored for time series data, including physiological signals; (3) Semi-supervised finetuning (SSFT): 1% labels of the training split of target domain data is available; (4) MME [41] and CDAC [28], two domain adaptation methods. Similar to (3), only 1% target domain labels are utilized. Fig. 4 shows the relative improvement in relative gains ($\Delta\%$)

**Table 3: Median Performance Across In-House Datasets.** Top-1, Top-2, and Top-3 results are highlighted in red, blue, and green, respectively, within both supervised (SL) and self-supervised (SSL) groups. **FAPEX** demonstrates consistently strong performance, achieving top-1 TO 3 rankings on the majority of datasets and metrics, reflecting its generalization and adaptability. For detailed results and statistical analysis, refer to App. C.

| Model | AGS | | | ATLE | | | IESS | | | KAIME | | | PCS | | |
|---|---|---|---|---|---|---|---|---|---|---|---|---|---|---|---|
| | SEN | F1 | ROC | SEN | F1 | ROC | SEN | F1 | ROC | SEN | F1 | ROC | SEN | F1 | ROC |
| ModernTCN | 87.0 | 85.0 | 93.2 | 91.7 | 90.2 | 100.0 | 73.4 | 73.4 | 67.2 | 83.4 | 73.2 | 87.3 | 85.9 | 85.4 | 86.3 |
| MRConv | 91.3 | 90.3 | 95.2 | 86.6 | 96.1 | 100.0 | 68.8 | 68.7 | 66.9 | 81.1 | 68.5 | 85.0 | 83.0 | 84.1 | 83.7 |
| MultiresNet | 90.1 | 88.8 | 96.1 | 85.4 | 84.3 | 100.0 | 72.1 | 70.4 | 68.7 | 80.4 | 63.7 | 82.5 | 69.2 | 64.4 | 83.9 |
| Omni-Scale | 91.7 | 90.9 | 95.2 | 87.8 | 98.6 | 99.9 | 67.9 | 68.7 | 67.2 | 81.0 | 68.8 | 83.0 | 80.0 | 79.6 | 80.9 |
| SPaRCNet | 89.1 | 87.5 | 93.4 | 84.0 | 81.7 | 99.8 | 60.7 | 64.9 | 61.4 | 82.0 | 77.1 | 86.5 | 85.5 | 84.4 | 91.0 |
| EEGConformer | 89.8 | 88.5 | 94.4 | 88.5 | 91.2 | 100.0 | 66.1 | 67.9 | 67.0 | 81.4 | 73.4 | 87.1 | 77.1 | 78.8 | 84.3 |
| EEGMamba | 93.8 | 93.5 | 96.8 | 88.2 | 85.0 | 100.0 | 69.6 | 70.0 | 68.8 | 80.4 | 69.4 | 83.4 | 70.8 | 73.3 | 85.6 |
| iTransformer | 89.5 | 87.8 | 95.3 | 54.9 | 2.9 | 99.8 | 53.4 | 54.5 | 66.4 | 81.3 | 63.4 | 87.1 | 74.3 | 73.0 | 83.4 |
| Nonformer | 93.2 | 92.7 | 96.7 | 84.7 | 97.5 | 99.8 | 69.7 | 74.7 | 68.9 | 79.5 | 90.3 | 81.6 | 68.8 | 63.7 | 84.1 |
| PatchTST | 90.5 | 89.3 | 95.5 | 86.6 | 93.0 | 100.0 | 61.6 | 63.8 | 67.5 | 83.0 | 73.8 | 88.5 | 71.9 | 71.2 | 73.8 |
| Pathformer | 92.5 | 91.8 | 96.7 | 88.7 | 95.3 | 100.0 | 71.3 | 72.1 | 68.6 | 80.6 | 67.4 | 85.3 | 78.7 | 80.9 | 83.7 |
| SeizureFormer | 92.1 | 91.3 | 95.4 | 86.3 | 97.9 | 99.9 | 69.7 | 69.9 | 66.7 | 77.3 | 53.4 | 85.9 | 58.6 | 62.4 | 59.7 |
| ATFNet | 85.2 | 84.1 | 90.8 | 83.1 | 97.0 | 99.8 | 59.7 | 56.2 | 68.2 | 65.0 | 45.7 | 71.9 | 74.7 | 73.5 | 84.6 |
| FreTS | 88.7 | 87.0 | 93.0 | 70.2 | 70.8 | 77.8 | 42.7 | 32.8 | 67.0 | 54.3 | 56.4 | 73.8 | 70.5 | 72.7 | 77.8 |
| NFM | 88.7 | 87.0 | 93.0 | 71.3 | 71.8 | 81.7 | 52.4 | 56.2 | 61.2 | 76.8 | 64.4 | 80.4 | 73.6 | 76.9 | 83.0 |
| TSLANet | 94.4 | 94.2 | 97.3 | 91.4 | 91.6 | 100.0 | 73.8 | 72.9 | 66.2 | 82.2 | 88.4 | 82.4 | 84.6 | 84.0 | 84.9 |
| AdaWaveNet | 89.0 | 87.6 | 95.1 | 82.8 | 96.6 | 99.8 | 70.0 | 70.7 | 66.3 | 70.5 | 77.7 | 84.8 | 72.4 | 74.2 | 81.3 |
| Medformer | 88.7 | 88.0 | 96.1 | 88.2 | 98.9 | 99.9 | 73.7 | 73.1 | 66.9 | 73.2 | 45.2 | 72.3 | 77.9 | 77.1 | 96.3 |
| MTST | 91.6 | 90.0 | 98.0 | 84.1 | 97.2 | 99.8 | 60.3 | 56.4 | 70.1 | 59.0 | 60.2 | 74.2 | 72.8 | 70.3 | 74.1 |
| Pyraformer | 92.0 | 91.3 | 96.5 | 85.1 | 97.7 | 99.8 | 71.4 | 70.2 | 66.9 | 83.2 | 56.6 | 86.4 | 76.8 | 79.9 | 82.3 |
| SimpleTM | 85.1 | 83.5 | 88.2 | 90.8 | 90.4 | 99.9 | 66.7 | 68.8 | 64.4 | 80.0 | 81.1 | 84.6 | 76.0 | 75.0 | 84.5 |
| TimesNet | 89.7 | 88.3 | 94.3 | 82.1 | 96.3 | 99.8 | 59.9 | 63.6 | 66.1 | 80.1 | 81.1 | 84.6 | 77.7 | 81.0 | 84.6 |
| TimeMixer | 92.3 | 91.6 | 96.6 | 87.9 | 95.3 | 99.8 | 71.7 | 71.0 | 68.6 | 82.1 | 90.0 | 85.2 | 81.1 | 83.5 | 85.4 |
| **FAPEX-Small (SL)** | 94.1 | 93.7 | 98.4 | 87.2 | 98.4 | 99.9 | 70.8 | 70.4 | 71.7 | 86.9 | 92.1 | 89.3 | 81.0 | 81.2 | 94.1 |
| **FAPEX-Base (SL)** | 94.9 | 94.6 | 99.5 | 88.0 | 98.8 | 99.9 | 72.3 | 72.4 | 71.4 | 87.0 | 95.6 | 90.1 | 91.5 | 91.5 | 96.3 |
| Brant | 93.2 | 92.7 | 96.6 | 87.9 | 83.0 | 99.9 | 68.0 | 67.7 | 69.5 | 74.5 | 74.8 | 74.4 | 83.1 | 82.3 | 95.8 |
| CBraMod | 90.8 | 90.6 | 98.7 | 87.9 | 82.8 | 99.9 | 79.6 | 80.7 | 76.2 | 81.0 | 79.8 | 83.7 | 81.1 | 83.5 | 85.4 |
| EEGPT | 93.4 | 92.9 | 98.5 | 88.2 | 83.2 | 100.0 | 74.2 | 73.9 | 71.4 | 78.4 | 77.3 | 78.6 | 85.5 | 84.4 | 91.0 |
| Neuro-BERT | 93.7 | 93.2 | 96.8 | 83.3 | 90.8 | 100.0 | 75.3 | 75.0 | 71.5 | 81.7 | 80.7 | 83.6 | 80.8 | 81.9 | 96.8 |
| VQ_MTM | 87.9 | 85.5 | 94.6 | 81.7 | 79.5 | 99.9 | 72.8 | 72.9- | 69.8 | 62.8 | 64.7 | 78.2 | 81.0 | 81.2 | 94.1 |
| COMETS | 93.9 | 93.4 | 98.2 | 87.7 | 83.2 | 99.8 | 67.6 | 68.2 | 79.3 | 80.6 | 80.1 | 84.0 | 80.8 | 81.9 | 96.8 |
| MF-CLR | 91.2 | 90.0 | 97.0 | 84.4 | 82.8 | 100.0 | 79.7 | 80.8 | 75.6 | 80.9 | 79.8 | 86.3 | 79.2 | 77.4 | 97.2 |
| TS2Vec | 94.7 | 94.5 | 97.5 | 62.7 | 76.1 | 73.6 | 72.4 | 73.6 | 72.9 | 76.1 | 76.2 | 76.7 | 69.0 | 65.9 | 96.4 |
| **FAPEX-Small (SSL)** | 94.1 | 93.7 | 98.4 | 94.0 | 92.8 | 100.0 | 81.5 | 83.7 | 83.4 | 87.4 | 87.1 | 89.3 | 91.0 | 91.0 | 96.7 |
| **FAPEX-Base (SSL)** | 95.2 | 94.9 | 99.7 | 94.8 | 98.0 | 100.0 | 83.7 | 84.9 | 85.9 | 88.7 | 88.4 | 91.4 | 95.0 | 95.0 | 97.5 |

of `FAPEX`-Base over `Neuro-BERT` and `CBraMod` in median F1. `FAPEX`-Base consistently achieves positive Δ% in F1 across diverse cases. It excels in the SOT setup, with Δ% often exceeding 30%, highlighting its strong generalization without target supervision relative to other models. In more informative setups like CDAC and MME, where SOTA models improve with target data, `FAPEX`-Base still outperforms or matches them in most cases, despite the narrowing gap and occasional dataset-specific underperformance. This resilience underscores its robust architecture and clinical potential in label-scarce settings and adaptivity to different finetuning techniques. See App. C for full results.

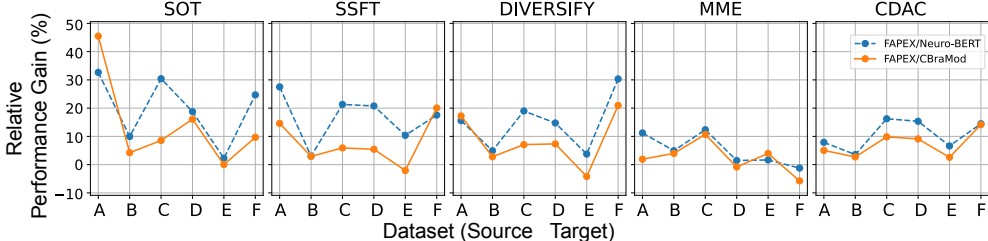

**Figure 4: Relative improvement in F1-score medians (Δ%) of `FAPEX`-Base over `Neuro-BERT` and `CBraMod` across five distinct transfer learning setups for six source-target dataset pairs.** `FAPEX`-Base demonstrates consistent performance gains for most cases, under both weak (SOT) and stronger supervision regimes (CDAC). A: KAIME → AGS, B: AGS → BEIRUT, C: IESS → BEIRUT, D: LPIRE → AGS, E: LPIRE → IESS, F: LPIRE → KAIME). `FAPEX`-Base consistently achieves superior performance.

**Ablation study and further analysis (RQ4).** To evaluate the contributions of each component within `FAPEX`, we conduct comprehensive ablation experiments. These studies isolate the effects of core modules—FrNFO, APCE, and SCA—on seizure prediction performance, providing insights into their individual and collective impacts (see App. D). We further explored the representational

characteristics and interpretability of FAPEX (see App. B). These analyses offer deeper insights into the model's decision-making processes and its alignment with known neural patterns.

## 4    Conclusion

We presennt FAPEX, a compact yet powerful neural architecture that integrates fractional frame theory directly into its core operators. Unlike the trend toward ever-larger models, FAPEX strategically leverages fractional neural frame operators to jointly encode amplitude and phase, achieving provable robustness against deformation and superior preservation of high-frequency biomarkers essential for precise seizure prediction. Extensive evaluations across fully supervised, self-supervised, and multi-cohort, multi-species out-of-domain settings consistently demonstrate that FAPEX surpasses specialized baselines and even large foundation models under comparable data regimes. These results establish FAPEX as a significant step forward in AI for healthcare with strong potential for improving clinical epilepsy management. Future work will aim to expand clinical datasets through collaboration with medical centers, incorporate complementary neuroimaging modalities, and explore deployment on wearable devices and closed-loop neurostimulation systems. Additionally, further theoretical analysis of phase–amplitude disentanglement and interpretability will be prioritized to enhance clinical trust and impact.

## 5    Acknowledgments

This work was supported by the Science and Technology Innovation 2030 - Brain Science and Brain-Inspired Intelligence Project (Grant No. 2021ZD0201301), the National Key Research and Development Program of China under Contract (Grant No. 2024YFA1610900), the National Natural Science Foundation of China (Grant Nos. 9257020, U20A20221, 12147101), and the Shanghai Municipal Science and Technology Committee of Shanghai Outstanding Academic Leaders Plan (Grant No. 21XD1400400). We thank the Shanghai Institute for Mathematics and Interdisciplinary Sciences (SIMIS)

for financial support (Grant No. SIMIS-ID-2025-NC). The computations were performed on the CFFF platform of Fudan University.

## Reference

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
