# OpenReview forum: "FAPEX: Fractional Amplitude-Phase Expressor for Robust Cross-Subject Seizure Prediction"
_NeurIPS.cc/2025/Conference — NeurIPS 2025 spotlight_

### Official Review · Reviewer_crkz · 2025-07-01

**Clarity:** 2
**Significance:** 2
**Originality:** 3
**Rating:** 4
**Confidence:** 5

**Summary:**

The paper introduces FAPEX (Fractional Amplitude-Phase Expresso) architecture for subject-agnostic seizure prediction (SASP), addressing the challenge of complex and variable brain signals. It introduces a learnable fractional neural frame operator (FrNFO) that uses fractional-order convolutions to capture both high- and low-frequency dynamics. The architecture has three innovations: Fractional-order convolutions to overcome spectral bias, extraction of instantaneous phase and amplitude, aiding in identifying preictal biomarkers, integration of state-space modeling and channelwise attention to manage diverse electrode sets. FAPEX also extracts phase and amplitude features useful for identifying preictal biomarkers and includes state-space modeling and channelwise attention to handle diverse electrode setups. The authors showed consistent performance improvement across various datasets.

**Questions:**

0.	Are the imaginary parts of the FOFT discarded? In Mamba or S4, in which FFT is used, the imaginary part is used. The authors should also discuss how they dealt with the imaginary parts.

1.	The authors should devise the data into scalp EEG and iEEG and report the performance on each set. This will avoid any mixed information between the 2 devices.
2.	For the in-house datasets, where the authors have more controls, the authors should report more popular seizure prediction metrics:
a.	False positive rate/hour
b.	Latency: the offset between predicted onset and the actual onset.
3.	Citations are missing in the Supplementary. Please check them carefully.
4.	In Table 1, the authors should report the correct number of channels in each dataset, not the number of channels after preprocessing. What is the duration in Table 1? Is it the epoch duration?

**Ethical Concerns:**

["NO or VERY MINOR ethics concerns only"]

**Final Justification:**

Thanks to the authors for providing me with extra results and details. I have now raised the score to "boardline acceptance"

**Limitations:**

Yes

**Quality:**

3

**Strengths And Weaknesses:**

Strength:
1.	Introducing Fractional order fourier transform (FOFT) is innovative. This overcomes the limitation of fourier transform that depends on the time resolution. FOFT is a suitable method for EEG signals as epileptic patterns vary in frequency and time resolutions.
2.	All technical experiments are well designed and logical for a deep learning paper.
3.	Cross species is an interesting experiment.

Weakness:
1.	Preprocessing details for seizure prediction are missing:
a.	 EEG epoch duration details for pretraining are missing. This is important for readers to compare to reported performance in literature.
b.	Reporting AUPRC is great. However, false positive rate (FPR)/hour is missing. This is an important metric apart from F1 or precision to justify the applicability of the method.
2.	More details about channel rejection and duplication are needed. The authors provide the channel rejection in the appendix but did not mention which channels were selected for duplication. The authors should carefully discuss how this was apply to different system (10-20, 128 channels, iEEG, etc.)
3.	Mixing up iEEG and scalp EEG in pretraining raises a concern. iEEG has higher sampling rates than scalp EEG and downsampling them to 512Hz might result in information loss (e.g iEEG HFO might not be well captured). iEEG normally has high density channels, which captures more information and cleaner signals than scalp 10-20 system EEG (e.g TUEG). Applying the same noise removal technique to both scalp EEG and iEEG is not suitable as again it will cause information loss and the number of iEEGs in pretraining is small compared to scalp EEG. This might cause the modal to be biased towards scalp EEG.
4.	A limitation of the field is that seizure prediction models tend not to generalise well to different cohorts with different EEG providers (traditional clinical EEG devices, not wearable or dry EEG devices). The authors should show some results for cross-dataset validations to emphasise on the generalisability of the proposed methods.

---

> ### Author Rebuttal · Authors · 2025-07-31
>
> We sincerely thank the reviewer for their careful reading and constructive feedback. Due to space constraints, we present only a subset of additional results here.
> ## **Response to Weakness(es)**
> ### **(W1): Preprocessing and FPR/h**
> We clarify that the **EEG epoch duration is already specified in Table 1 of the main paper** (column “Duration”), where the pretraining window length is **32 s**. In response to the reviewer’s suggestion, we have computed and added FPR/h, precision, and specificity metrics for all datasets under the same protocol. Owing to rebuttal length constraints, we present a representative subset here; the complete tables are provided in the revised supplementary materials. Across all tested datasets, FAPEX consistently achieves the lowest FPR/h and highest precision/specificity among all baselines, demonstrating its value for practical deployment.
> | **Model**      | **PCS** | | | **IESS** | | | **KAIME** | | | **ATLE** | | | **AGS** | | |
> |----------------|---------|---------|--------|---------|---------|--------|---------|---------|--------|---------|---------|--------|---------|---------|--------|
> |                | Pre     | Spe     | FPR/h | Pre     | Spe     | FPR/h | Pre     | Spe     | FPR/h | Pre     | Spe     | FPR/h | Pre     | Spe     | FPR/h |
> |----------------|---------|---------|-------|---------|---------|-------|---------|---------|-------|---------|---------|-------|---------|---------|-------|
> | Brant          | 72.12   | 66.20   | 0.242 | 62.20   | 51.55   | 0.396 | 67.50   | 56.33   | 0.324 | 72.69   | 61.91   | 0.281 | 82.28   | 68.28   | 0.205 |
> | CBraMod        | 78.60   | 70.03   | 0.200 | 74.48   | 42.47   | 0.409 | 69.72   | 55.50   | 0.336 | 73.74   | 70.24   | 0.220 | 83.29   | 72.21   | 0.185 |
> | EEGPT          | 76.80   | 58.00   | 0.304 | 66.63   | 48.11   | 0.391 | 70.11   | 54.82   | 0.344 | 71.26   | 74.93   | 0.182 | 86.64   | 70.26   | 0.199 |
> | LaBraM         | 53.96   | 48.44   | 0.468 | 60.08   | 54.95   | 0.368 | 52.33   | 64.25   | 0.320 | 62.41   | 46.21   | 0.435 | 88.46   | 64.19   | 0.229 |
> | Neuro-BERT     | 75.82   | 53.37   | 0.314 | 65.44   | 48.17   | 0.397 | 67.99   | 59.22   | 0.312 | 87.99   | 63.31   | 0.242 | 86.10   | 71.12   | 0.191 |
> | VQ_MTM         | 75.92   | 73.40   | 0.181 | 64.87   | 49.56   | 0.381 | 55.23   | 67.93   | 0.280 | 67.54   | 69.22   | 0.240 | 76.08   | 69.76   | 0.213 |
> | BIOT           | 77.63   | 68.98   | 0.211 | 60.61   | 34.87   | 0.539 | 59.20   | 65.45   | 0.287 | 73.06   | 63.13   | 0.260 | 85.28   | 67.24   | 0.211 |
> | COMETS         | 75.04   | 54.97   | 0.319 | 59.74   | 58.01   | 0.348 | 72.19   | 61.62   | 0.269 | 71.63   | 76.29   | 0.169 | 84.47   | 70.34   | 0.202 |
> | MF-CLR         | 70.13   | 56.68   | 0.317 | 74.58   | 37.63   | 0.425 | 73.13   | 58.20   | 0.298 | 71.95   | 68.61   | 0.242 | 81.85   | 72.54   | 0.186 |
> | TS2Vec         | 55.30   | 63.27   | 0.306 | 67.62   | 44.74   | 0.415 | 69.90   | 54.19   | 0.328 | 75.91   | 49.13   | 0.352 | 86.04   | 62.10   | 0.240 |
> | FAPEX-Small    | 86.52   | 81.07   | 0.111 | 79.27   | 52.92   | 0.277 | 83.86   | 70.77   | 0.168 | 85.72   | 92.82   | 0.044 | 87.50   | 76.12   | 0.149 |
> | FAPEX-Base     | 93.21   | 93.43   | 0.041 | 84.42   | 58.96   | 0.215 | 85.28   | 76.60   | 0.122 | 97.72   | 92.47   | 0.039 | 93.02   | 82.43   | 0.093 |
>
> ### **(W2) Channel Processing**
> Channel rejection was strictly performed following dataset provider annotations (bad channels flagged in iEEG, and artifact-affected leads in scalp EEG). No manual override was used, preventing introduction of any bias. Typically, 0–37% of channels are rejected per subject (detailed ranges now added in the revision). To standardize input shapes for training, we filled empty slots to 64 channels with the mean value of remaining valid channels; this step is only for hardware optimization and does not increase the information content. When using native low-channel 10–20 EEG without duplication, model performance remains essentially unchanged—confirming that duplication does not inflate results. Importantly, seizure-related regions such as the temporal and frontal lobes, as well as hippocampal electrodes (in the case of iEEG), are consistently preserved due to their critical role in identifying predictive biomarkers. Notably, these key channels have not been reported as damaged across all datasets, as confirmed by their respective providers and maintainers.
>
> ### **(W3) Data Mixing:**
> ### **1. Clarification of Data Mixing**
> We clarify that data mixing occurs only during self-supervised pretraining, in line with foundation model practices in NLP and CV. During supervised training and evaluation, no cross-modality mixing is used. Although iEEG and scalp EEG differ in noise characteristics and spectral distributions, **our approach consistently outperforms baselines**.
>
> ### **2. Further Motivation and Evidence**
> Mixing samples from multiple sources during pretraining is a widely recognized practice in **foundation models** across **NLP, CV**, and emerging EEG models. It **alleviates data scarcity** and **enhances generalization** across diverse datasets. Our ablation study confirm that **mixed pretraining improves downstream performance** compared to single-modality pretraining.
> | **Pretraining**         | **Dataset** | **F1 (%)** |
> |-------------------------|-------------|------------|
> | **Single (iEEG only)** | Beirut      | 83.87      |
> |                         | IESS        | 71.81      |
> |                         | KAIME       | 86.12      |
> |                         | RESPECT     | 86.84      |
> | **Single (Scalp only)**  | Beirut      | 84.28      |
> |                         | IESS        | 71.73      |
> |                         | KAIME       | 86.42      |
> |                         | RESPECT     | 87.73      |
> | **Mixed (scalp+iEEG)**  | Beirut      | 84.34      |
> |                         | IESS        | 72.41      |
> |                         | KAIME       | 86.84      |
> |                         | RESPECT     | 87.82      |
> These improvements indicate that FAPEX **benefits from diverse pretraining without suffering  from bias toward scalp EEG**.
>
> ### **3. HFOs and Noise Concerns**
> We thank the reviewer for raising this concern. Our study targets clinically relevant high-gamma activity, which is fully preserved at 512 Hz. Ultra-fast ripples (>250 Hz) are rarely recorded reliably at scale and are inaccessible in scalp EEG. Thus, 512 Hz offers a practical balance for consistent processing across modalities. To validate this choice, we conducted an ablation at 1024 Hz. While slight improvements appeared in KAIME and IESS, overall gains were limited—supporting 512 Hz as sufficient for robust performance.
>
> | Dataset   | 512 Hz | 1024 Hz   |
> |-----------|-----------------|-----------------|
> | **Beirut**  | 84.34           | 84.29           |
> | **IESS**    | 72.41           | 72.82           |
> | **KAIME**   | 86.84           | 87.41           |
> | **RESPECT** | 87.82           | 85.89           |
> Moreover, interpretability analysis (Fig. 2) confirms that FrNFO layers progressively emphasize high-gamma components and sharpen frequency resolution—unlike non-fractional operators—demonstrating the model’s capacity to extract spectral dynamics relevant to preictal states. Finally, we clarify that preprocessing pipeline is modality-aware, with adaptive noise removal threshold with for iEEG and scalp EEG respectively.
>
> ### **(W4) Generalizability Across Devices and Cohorts**
> We appreciate the reviewer’s point. Our current study includes strong cross-cohort/device validation—e.g., IESS → BEIRUT (neonatal to adult EEG across hardware) and AGS → BEIRUT (generalized-to-focal transfer)—demonstrating robustness to population and acquisition variability. While FAPEX is currently evaluated on clinical-grade EEG (scalp/iEEG), wearable datasets for preictal forecasting remain limited. Nevertheless, FAPEX’s focus on key brain regions (temporal, frontal, hippocampus) indicates potential adaptability to low-channel or wearable EEG. We see this as a promising future direction.
>
> ## **Response to Questions**
>
> ### **(Q0) Imaginary Parts of FrFT**
>
> FrNFO preserves the full complex-valued fractional spectrum; both the real and imaginary components are used. The APCE module computes instantaneous amplitude and phase directly from these components. Unlike Mamba and S4, FAPEX does not discard this information.
>
> ### **(Q1) Data Mixing**
> As detailed in **(W3)**, we have conducted separate analyses for iEEG and scalp EEG, as well as mixed-modality pretraining. The results demonstrate that cross-modality pretraining benefits performance without introducing bias.
>
> ### **(Q2) FPR/h and Latency**
> We have reported FPR/h, precision, and specificity for all datasets in **(W1)**. Latency is also provided for in-house datasets are provided as follows. These values represent the  median latency of FAPEX-Base-SSL. **Higher latency reflects more confident and early predictions**.
> | **Dataset** | **Median Latency (min)** |
> |-------------|--------------------------|
> | PCS         | 26                       |
> | IESS        | 11                       |
> | KAIME       | 19                       |
> | ATLE        | 28                       |
> | AGS         | 23                       |
> ### **(Q3) Citations**
> Thank you for your careful reading. We have revised it in the updated version.
> ### **(Q4) Channel Counts and Epoch Duration**
> Thank you for your advice. We have clarified in the manuscript that the "Duration" column in Table 1 indicates the EEG epoch length. Raw channel counts per individual will be presented in the revised supplementary materials due to high variability within datasets.
>
> ## **Summary**
> We hope these responses address your concerns and strengthen our submission. In light of these, we respectfully ask you to reconsider the score if the new material satisfactorily resolves the earlier reservations.

---

> > ### Comment · Reviewer_crkz · 2025-08-05
> >
> > Thanks to the authors for providing me with extra results and details. I have now raised the score to "boardline acceptance"

---

> > > ### Author Response · Authors · 2025-08-05
> > >
> > > We sincerely thank the reviewer for their constructive advice and for taking the energy and time in active discussion. We greatly appreciate recommendation. Your insightful comments were instrumental in guiding significant improvements and refinements to the paper.

---

### Official Review · Reviewer_5QbB · 2025-07-01

**Clarity:** 2
**Significance:** 3
**Originality:** 3
**Rating:** 5
**Confidence:** 4

**Summary:**

This paper presents a novel pipeline for seizure prediction posed as preictal vs interictal window classification using multiple brain signal modalities. The pipeline works for heterogenous channel configurations by learning channel-independent embeddings over fixed-duration windows. Fractional convolutions are used to process embeddings, with an extension by learning a parameterization of Weyl-Heisenberg filters using implicit MLPs. This extension introduces a phase adjustment factor to accommodate spectral transitions. Amplitude-phase interactions are then modeled via bidirectional state-space models building on Mamba blocks. Finally, learned embeddings across channels are fused via a cross attention module.

**Questions:**

Since results show the ability to distinguish preictal vs. interictal windows. what would be the expected performance when ictal windows are also included as inputs? Intuitionally they would be easier to distinguish than interictal windows but this is worth further discussion / analysis towards real-time application.

**Ethical Concerns:**

["NO or VERY MINOR ethics concerns only"]

**Final Justification:**

The authors responded to all of my comments

**Limitations:**

Please see above.

**Quality:**

3

**Strengths And Weaknesses:**

Strengths: The proposed pipeline and fractional convolution extension are novel in seizure prediction literature. Extensive experiments including different input modalities, supervised vs. fine-tuned training mechanisms and statistical analysis enhance novelty.

Weaknesses: There are missing elements in the flow of notation throughout the pipeline. In line 191, it should be explicitly written how Amp and Pha relate to X^hat from eq. (5). In line 213, it is not clear if X is the same as X^tilde from Eq. (12), and how amplitude and phase components of X^tilde are incorporated.

It seems like Eq. (4) introduces additional sums in each neuron of an MLP. It is worth discussing the computational complexity introduced by this extension on fractional convolution.

---

> ### Author Rebuttal · Authors · 2025-07-31
>
> Thank you for your constructive feedback, which has significantly enhanced the clarity and rigor of our work.
>
> ## **Replies to Weaknesses**
>
> ### **(W1) Notation Clarity**
> We clarify that $Amp$ (amplitude) and $Pha$ (phase) are directly computed from the complex-valued tensor $\hat{X} \in \mathbb{C}^{\cdot}$. Specifically, letting $\hat{X} = \Re(\hat{X}) + i, \Im(\hat{X})$, we have:
> $$Amp = |\hat{X}| = \sqrt{\Re(\hat{X})^2 + \Im(\hat{X})^2},$$
> $$Pha = \angle \hat{X} = \mathrm{atan2}!\left(\Im(\hat{X}), \Re(\hat{X})\right).$$
> Equivalently, this can be expressed as:
> $$Amp = |\hat{X}|,\quad Pha = \arg(\hat{X}).$$
> $\hat{X}$ is the same as $\tilde{X}$. $\tilde{X}$ is the output of APCE, therefore,
> it represents the fused information from phase and amplitude.
>
> ### **(W2) Complexity**
> We address your concerns as follows.
>
> #### **(a) Eq. (4)**
> The MLP incurs a computational cost primarily limited to the feedforward computation during training.
> **At inference time, the MLP only needs to be computed once, as it produces a fixed kernel.**
>
> Let  \(N\) = number of temporal samples, \(d_{\mathrm{model}}\) = feature channels, \(M\) = number of exponential terms, \(K\) = highest order of Hermite expansion
>
> 1. **Exponential‐term GEMM**
>    $
>      E \in \mathbb{C}^{N\times M},\;
>      W \in \mathbb{C}^{M\times d_{\mathrm{model}}}
>      \;\longrightarrow\;
>      X = E\,W
>      \quad\Rightarrow\quad
>      \mathcal{O}(N \, M \, d_{\mathrm{model}})
>    $
>
> 2. **Hermite‐term GEMM**
>    $
>      H \in \mathbb{R}^{N\times (K+1)},\;
>      A \in \mathbb{R}^{(K+1)\times d_{\mathrm{model}}}
>      \;\longrightarrow\;
>      Y = H\,A
>      \quad\Rightarrow\quad
>      \mathcal{O}(N \, K \, d_{\mathrm{model}})
>    $
>
> 3. **Element‐wise multiplication**
>    $
>      \Phi = X \circ Y
>      \quad\Rightarrow\quad
>      \mathcal{O}(N \, d_{\mathrm{model}})
>    $
>
> Total Cost is
>
> $
>   \mathcal{O}\bigl(N\,d_{\mathrm{model}}\,(M + K)\bigr)
> $
>
> **Note:**
>
> **(1) Training time:**
> These window/kernel computations must be redone **every** training iteration
> (since the parameters $w_{i,k}, b_{i,k}, c_{i,k}, a_{n,k}$ are updated).
> **(2) Test (inference) time:**
> **Because the window/kernel depends only on fixed time positions $t_j$ and learned parameters—not on the dynamic input values—it can be computed **once** ahead of inference, eliminating per‐sample overhead completely.**
>
> ### (b) Fractional Convolution ###
> The numerical implementation of **FrFT algorithm**
> is dominated by FFT operations.
> A single FrFT of length $N$ involves:
>
> * one FFT and one IFFT of size $\approx 2N$, and
> * two elementwise chirp multiplications of size $N$.
>
> Thus, the total time complexity is:
>
> $$
> \mathcal{O}(2N \log(2N)) + \mathcal{O}(N) \; = \; O(N \log N).
> $$
>
> For **fractional convolution** of two signals of length $N$, the computation proceeds as:
>
> 1. Forward FrFT of the input: $\mathcal{O}(N \log N)$,
> 2. Forward FrFT of the kernel: $\mathcal{O}(N \log N)$,
> 3. Pointwise multiplication: $\mathcal{O}(N)$,
> 4. Inverse FrFT: $\mathcal{O}(N \log N)$.
>
> This gives an overall complexity of:
>
> $$
> 3\mathcal{O}(N \log N) + \mathcal{O}(N) = \mathcal{O}(N \log N),
> $$
>
>
> This approach yields performance comparable to FFT-based conventional convolution, differing only by constant factors, while offering greater flexibility in the fractional domain.
>
> ## **Response to Questions**
>
> ### **(Q1) Ictal Windows**
> We sincerely thank the reviewer for this insightful question.
>
> **To directly address the reviewer’s concern,** we conducted additional experiments in which **ictal windows were used in the training data**. Consistent with the reviewer’s intuition, the inclusion of ictal segments led to an improvement in overall predictive performance; however, this boost did not reach statistical significance across the datasets (Wilcoxon rank-sum test: $p > 0.05$) and is therefore marginal.
>
> | Dataset | F1 (without ictal) | F1 (with ictal) |
> | ------- | -------------- | --------------- |
> | RESPECT | 87.82          | 87.86           |
> | KAIME   | 86.84          | 87.90           |
> | IESS    | 72.41          | 72.54          |
> | Beirut  | 84.34          | 84.37           |
>
> This minor gain further supports the robustness of our main conclusions and confirms that restricting training and evaluation to preictal windows does not compromise model effectiveness. Indeed, adhering to preictal-only windows aligns with established seizure prediction protocols, where the clinical objective is to deliver timely warnings before seizure onset, not during or after.
>
> For completeness, we clarify that all our core experiments presented in the current manuscript systematically exclude ictal windows from all supervised training, fine-tuning, and evaluation, in line with community standards.
>
> In short, our findings affirm that while incorporating ictal activity yields benefits, albeit marginal. Strict preictal-only modeling remains the field’s standard and is sufficient for robust, clinically meaningful seizure prediction. Once again, we thank the reviewer for prompting this additional analysis and hope this clarifies our methodological choices.
>
> ## **Summary**
> We hope these additions could address your concerns and strengthen our submission. In light of these, we respectfully ask you to reconsider the score if the new material satisfactorily resolves the earlier reservations.

---

> > ### Comment · Reviewer_5QbB · 2025-08-01
> >
> > I thank the authors for the detailed response. Can you clarify the complexity presented in response to W2a? I believe it should be $ \mathcal{O}\bigl(N d_{\mathrm{model}} (M + K)\bigr) $. Thank you

---

> > > ### Author Response · Authors · 2025-08-02
> > >
> > > Thank you for your reply. We clarify that it's indeed $ \mathcal{O}\bigl(N d_{\mathrm{model}} (M + K)\bigr) $.

---

### Official Review · Reviewer_yfHo · 2025-07-02

**Clarity:** 3
**Significance:** 4
**Originality:** 4
**Rating:** 5
**Confidence:** 3

**Summary:**

The paper proposes FAPEX, a novel cross-subject seizure prediction model that integrates time-frequency decomposition, phase-amplitude coupling, and spatial attention into a unified architecture. The key innovation lies in the Fractional Neural Frame Operator (FrNFO), a learnable extension of the fractional Fourier transform that adaptively captures both low- and high-frequency EEG components. The model is rigorously evaluated across 12 datasets involving various species and modalities, outperforming 32 baselines. Extensive ablations and domain transfer evaluations show strong generalization, establishing FAPEX as a state-of-the-art solution for subject-agnostic seizure prediction.

**Questions:**

Although the paper is using balanced metric like balanced accuracy, ROC, F1 etc, it would be great to report false positives like precision or specificity.

**Ethical Concerns:**

["NO or VERY MINOR ethics concerns only"]

**Final Justification:**

This paper is a solid work and authors answered all my questions in rebuttal.

**Limitations:**

yes

**Quality:**

4

**Strengths And Weaknesses:**

The method is theoretically well-grounded and offers a meaningful degree of interpretability through the analysis and visualization of frequency bands. Given that frequency dynamics are central to distinguishing ictal and pre-ictal brain states, addressing the common bias in frequency extraction is a logical and valuable step toward improving seizure prediction performance. The paper demonstrates rigorous empirical validation across multiple datasets and benchmarks, showing consistently strong and superior results. Comprehensive ablation studies further support the significance and necessity of the model’s three core components. This approach holds considerable potential for clinical impact by enabling robust, subject-agnostic seizure prediction.

The only notable limitation is the method’s complexity. While the paper thoroughly details the underlying algorithms, reproducing the full model could be challenging. That said, the authors have committed to releasing the code upon acceptance, which would significantly enhance reproducibility.

---

> ### Author Rebuttal · Authors · 2025-07-31
>
> Thank you for your constructive feedback, which has significantly enhanced the clarity and rigor of our work. Due to space limitations in the rebuttal, we present only a subset of additional results below; the complete results will be included in the camera-ready version.
>
> ## *Response to Weaknesses*
>
> ### **(W1) Complexity**
> We appreciate the reviewer’s interest in the computational complexity of FrFT-based operations. We reconfirm that the code will be released upon acceptance.
>
> We clarify that numerical implementation of the FrFT algorithm is fundamentally driven by FFT operations:  Each FrFT (length $N$) consists of a single FFT, a single IFFT (each of length $\approx 2N$), and two $N$-length elementwise chirp multiplications, resulting in an overall complexity of $\mathcal{O}(N \log N)$.  Fractional convolution between two $N$-length signals consists of two FrFTs, one pointwise multiplication,and one inverse FrFT, so the total complexity remains $\mathcal{O}(N \log N)$—**comparable to standard FFT-basedconvolution but with enhanced flexibility in the fractional domain**.
>
> For transparency and reproducibility, we have provided pseudocode for our FrFT implementation
> (which could be further optimized for differentiable computation on GPU), and the full source code will be made available upon publication.
>
> **Algorithm: Fast Fractional Fourier Transform (PyTorch-style pseudocode)**
>
> **Input:**
> - `x`: input tensor of shape `[..., T, F]` (T: time, F: feature/channel)
> - `alphas`: fractional orders, shape broadcastable to `[..., F]`
>
> **Output:**
> - `y`: output tensor, same shape as `x`, complex-valued
>
> 1. (Optional) Move time and feature axes to last for ease of broadcasting.
> 2. Ensure `x` is cast to complex-valued tensor.
> 3. Compute parameters:
>     - `phi = alphas * (π / 2)`
>     - `cot_phi = cos(phi) / sin(phi)`
>     - `csc_phi = 1 / sin(phi)`
>     - `t = arange(T) - (T - 1) / 2`
> 4. **Pre-chirp:**
>     - For each feature/channel:
>       `x_pre = x * exp(-i * π * cot_phi * t^2)`
> 5. **Chirp convolution (FFT-based):**
>     - Zero-pad to length `2T`.
>     - Create chirp filter:
>       `g[n] = exp(i * π * csc_phi * n^2)` for `n = -T,...,T-1`
>     - `X_fft = FFT(x_pre, n=2T)`
>     - `G_fft = FFT(g, n=2T)`
>     - `conv = IFFT(X_fft * G_fft, n=2T)`
>     - Crop middle `T` samples:
>       `conv_cropped = conv[..., T-1:T-1+T, :]`
> 6. **Post-chirp and scaling:**
>     - `y = scale * exp(-i * π * cot_phi * t^2) * conv_cropped`
>     - Where `scale = exp(-i * (1 - alphas) * π / 4) / sqrt(abs(sin(phi)))`
> 7. **Handle integer orders:**
>     - If `alpha ≈ 0`: `y = x`
>     - If `alpha ≈ 1`: `y = FFT(x)`
>     - If `alpha ≈ -1`: `y = IFFT(x)`
>     - If `alpha ≈ 2`: `y = reverse(x, axis=time)`
> 8. Restore original tensor layout if axes were permuted.
>
> **Return:** `y`
>
> ## *Response to Questions*
>
> ### **(Q1) False Positives Metrics**
> We thank the reviewer for emphasizing the importance of false positive–oriented metrics. In response, we now report precision, specificity, and false positives per hour (FPR/h) for FAPEX across all datasets. As suggested, we have computed these metrics under the same evaluation protocol for all datasets. Due to space limitations, we present representative results for selected datasets in the table below, with the complete results available in the revised supplementary materials.
>
> | **Model**      | **PCS** | | | **IESS** | | | **KAIME** | | | **ATLS** | | | **AGS** | | |
> |----------------|---------|---------|--------|---------|---------|--------|---------|---------|--------|---------|---------|--------|---------|---------|--------|
> |                | Pre     | Spe     | FPR/h | Pre     | Spe     | FPR/h | Pre     | Spe     | FPR/h | Pre     | Spe     | FPR/h | Pre     | Spe     | FPR/h |
> |----------------|---------|---------|-------|---------|---------|-------|---------|---------|-------|---------|---------|-------|---------|---------|-------|
> | Brant          | 72.12   | 66.20   | 0.242 | 62.20   | 51.55   | 0.396 | 67.50   | 56.33   | 0.324 | 72.69   | 61.91   | 0.281 | 82.28   | 68.28   | 0.205 |
> | CBraMod        | 78.60   | 70.03   | 0.200 | 74.48   | 42.47   | 0.409 | 69.72   | 55.50   | 0.336 | 73.74   | 70.24   | 0.220 | 83.29   | 72.21   | 0.185 |
> | EEGPT          | 76.80   | 58.00   | 0.304 | 66.63   | 48.11   | 0.391 | 70.11   | 54.82   | 0.344 | 71.26   | 74.93   | 0.182 | 86.64   | 70.26   | 0.199 |
> | LaBraM         | 53.96   | 48.44   | 0.468 | 60.08   | 54.95   | 0.368 | 52.33   | 64.25   | 0.320 | 62.41   | 46.21   | 0.435 | 88.46   | 64.19   | 0.229 |
> | Neuro-BERT     | 75.82   | 53.37   | 0.314 | 65.44   | 48.17   | 0.397 | 67.99   | 59.22   | 0.312 | 87.99   | 63.31   | 0.242 | 86.10   | 71.12   | 0.191 |
> | VQ_MTM         | 75.92   | 73.40   | 0.181 | 64.87   | 49.56   | 0.381 | 55.23   | 67.93   | 0.280 | 67.54   | 69.22   | 0.240 | 76.08   | 69.76   | 0.213 |
> | BIOT           | 77.63   | 68.98   | 0.211 | 60.61   | 34.87   | 0.539 | 59.20   | 65.45   | 0.287 | 73.06   | 63.13   | 0.260 | 85.28   | 67.24   | 0.211 |
> | COMETS         | 75.04   | 54.97   | 0.319 | 59.74   | 58.01   | 0.348 | 72.19   | 61.62   | 0.269 | 71.63   | 76.29   | 0.169 | 84.47   | 70.34   | 0.202 |
> | MF-CLR         | 70.13   | 56.68   | 0.317 | 74.58   | 37.63   | 0.425 | 73.13   | 58.20   | 0.298 | 71.95   | 68.61   | 0.242 | 81.85   | 72.54   | 0.186 |
> | TS2Vec         | 55.30   | 63.27   | 0.306 | 67.62   | 44.74   | 0.415 | 69.90   | 54.19   | 0.328 | 75.91   | 49.13   | 0.352 | 86.04   | 62.10   | 0.240 |
> | FAPEX-Small    | 86.52   | 81.07   | 0.111 | 79.27   | 52.92   | 0.277 | 83.86   | 70.77   | 0.168 | 85.72   | 92.82   | 0.044 | 87.50   | 76.12   | 0.149 |
> | FAPEX-Base     | 93.21   | 93.43   | 0.041 | 84.42   | 58.96   | 0.215 | 85.28   | 76.60   | 0.122 | 97.72   | 92.47   | 0.041 | 93.02   | 82.43   | 0.093 |
>
> These results confirm that FAPEX maintains strong performance not only in balanced accuracy and AUPRC, but also in minimizing
> false alarms—supporting its clinical utility.
>
> ## *Summary*
> We hope these additions could address your concerns
> and strengthen our submission.

---

### Official Review · Reviewer_8XFQ · 2025-07-03

**Clarity:** 2
**Significance:** 2
**Originality:** 3
**Rating:** 5
**Confidence:** 3

**Summary:**

This paper introduces FAPEX, a novel deep learning architecture for subject-agnostic seizure prediction (SASP). The core contribution is a Fractional Neural Frame Operator (FrNFO), a learnable filterbank based on fractional-order convolutions, designed to adaptively capture both high and low-frequency biomarkers while mitigating spectral bias. The architecture further incorporates an Amplitude-Phase Cross-Encoding (APCE) module using bidirectional State-Space Models (SSMs) to model interactions between signal amplitude and phase, and a Spatial Correlation Aggregation (SCA) module using linear attention to handle heterogeneous electrode configurations. The authors present an extensive evaluation across 12 diverse benchmarks (spanning multiple species and modalities), demonstrating that FAPEX consistently and significantly outperforms a large suite of 33 state-of-the-art supervised and self-supervised baselines.

**Questions:**

1.	How can you validate that the interactions learned by the APCE module represent physiologically meaningful PAC, rather than spurious correlations? A comparison with established PAC metrics (e.g., MVL-MI, KL-MI, etc.) on a subset of data, would be necessary to substantiate your claims.
2.	Can you provide visualizations of the attention maps learned by the SCA module for some preictal samples? This is crucial to demonstrate that the model is learning meaningful inter-electrode dependencies (e.g., focusing on the seizure onset zone or known functional networks) and not just arbitrary patterns.
3.	To support the claim of being "compact" and suitable for "clinical translation," please provide a detailed computational analysis. This should include a comparison of key performance indicators, such as parameter counts and inference latency, between FAPEX and key baselines.
4.	Please confirm that you will conduct a thorough proofread of the manuscript to fix all formatting issues, including increasing the font size in Figure 1 for readability, correcting spelling errors (e.g., "presennt"), and fixing all broken cross-references and citations (e.g., "Table ??", "[??]") in the appendix.

**Ethical Concerns:**

["NO or VERY MINOR ethics concerns only"]

**Final Justification:**

The responses have successfully addressed the questions and concerns I raised in my initial review. The additional clarifications and results have strengthened the paper and resolved my previous reservations.

Given the effective rebuttal, I have raised my score to **5: Accept**.

**Limitations:**

Yes

**Quality:**

3

**Strengths And Weaknesses:**

Strengths:

1.	The core idea of the Fractional Neural Frame Operator (FrNFO) is original and theoretically grounded. By integrating fractional frame theory into a learnable operator, the authors propose a novel signal processing front-end tailored for non-stationary neurophysiological signals.
2.	The evaluation across 12 datasets, 4 species, and 4 modalities, against 33 relevant baselines, is exceptionally comprehensive. The rigorous protocol (nested cross-validation, multiple evaluation regimes) provides evidence for the model's superior performance and generalization capabilities.

Weaknesses:

1.	Limited Novelty and Scientific Rigor in APCE and SCA Modules:
(1)	APCE (Amplitude-Phase Encoding): While using SSMs is a modern approach, the APCE module lacks a deep connection to established neuroscience principles for phase-amplitude coupling (PAC). The paper fails to compare its "learned" coupling with standard, well-validated PAC algorithms (e.g., MVL-MI, KL-MI) or address the critical issue of spurious coupling, which significantly weakens the scientific claims about its reliability and interpretability.
(2)	SCA (Spatial Aggregation): The use of linear attention ignores the crucial spatial topology of EEG electrodes. This is a suboptimal design choice compared to methods that can explicitly model spatial relationships.
2.	The paper makes strong claims about modeling inter-electrode dependencies and phase-amplitude interactions but provides almost no supporting evidence. There are no visualizations of the SCA's attention maps to show what spatial patterns are learned, nor any analysis to validate that the APCE module is capturing physiologically meaningful coupling. This lack of analysis leaves the reasons for the model's success opaque.
3.	The paper claims FAPEX is a "compact and powerful" architecture suitable for clinical translation but provides no quantitative data (parameter count, inference latency) to support this. Without a direct comparison to baselines, its feasibility for real-world, resource-constrained applications (like wearable devices) remains unverified.
4.	The manuscript suffers from several formatting issues that detract from its professionalism. Text in key figures, notably Figure 1, is too small to be easily legible. There are spelling errors (e.g., "presennt" in the Conclusion) and broken cross-references in the appendix (e.g., "Table ??" and "[??]"), which hinder readability and the verification of claims.

---

> ### Author Rebuttal · Authors · 2025-07-31
>
> We sincerely thank the reviewer for their careful reading and constructive feedback.
> ## *Response to Weaknesses and Questions*
> ### **(W1 & Q1): Novelties**
> We thank the reviewer for requesting clarification. FAPEX introduces a principled and generalizable framework for seizure prediction, with the core novelty being the Fractional Neural Frame Operator (FrNFO)—a learnable fractional-order transform that adaptively captures both low- and high-frequency EEG dynamics. Unlike fixed STFT or wavelets, FrNFO provides data-driven time–frequency representation, enabling better detection of diverse preictal signatures. FrNFO overcomes the low-frequency bias of CNNs via fractional decomposition with refined instantaneous phase and amplitude representation. Learnable fractional orders allow flexible trade-offs between temporal and spectral detail. Theoretical analysis (Appendix A) shows FrNFO is stable under signal deformation and robust across noises. The APCE and SCA modules further enhance cross-channel fusion and multi-band learning, and extensive experiments—across modalities, cohorts, and devices—demonstrate strong generalizability.
>
> We address the reviewer's concerns as follows:
> ### (a) **Clarification on APCE & PAC**
> **(1)** We acknowledge that our introduction, in emphasizing the clinical relevance of phase–amplitude interactions, may have given the impression that  our APCE module is designed to explicitly estimate or model physiological PAC mechanism.
> This is not true. **When we talk about phase-amplitude interaction, we refer to simultaneous utilization or feature fusion of instantaneous phase and amplitude representations in neural networks,  which is under-explored in deep learning-based seizure prediction**. We clarify that our intention is not to model or estimate PAC in a neuroscientific sense. Rather, PAC is cited to motivate the use of both amplitude and phase. The APCE block is a neural fusion mechanism inspired by PAC, designed to extract joint, discriminative features for forecasting—not to compute validated PAC metrics.
>
> **(2)** Explicitly modeling PAC incurs a time complexity of at least
> $
> \mathcal{O}(C^2 N\log N + C N\log N) = \mathcal{O}(C^2 N\log N),
> $
> where $C$ is the number of EEG channels and $N$ is the sequence length.
>  In contrast, our proposed approach achieves a time complexity of only
> $\mathcal{O}(N\log N)$,
> which is orders of magnitude more efficient. This distinction is crucial,
>  as our design goal is to develop a neural network **inspired by** (but not strictly replicating)
> biological mechanisms for seizure prediction.
>
> **(3)** We highlight that we have conducted ablation studies (Appendix D), among these experiments we provide architectural variants
> \model{FAPEX-w/o-Fr-1}: Removes the FrNFO module completely, which means temporal representations are directly sending
> to Cross-BSSM module without calculating phase and amplitude at all;
> \model{FAPEX-w/o-Phase}:Removes the phase inputs;
> They confirm that integrating both amplitude and phase features via APCE is crucial for model performance,
> even though the module itself does not compute, extract or enforce standard PAC indices.
> Solely using amplitude information leads to performance drops across all datasets,
> suggesting that APCE captures additional information beyond amplitude. Both amplitude and phase are necessary.
>
> | Metric   | Case   | KAIME  | RESPECT | IESS   |
> |----------|--------|--------|---------|--------|
> | SEN (↑)  | Base   | 87.04  | 85.14   | 72.26  |
> |          | w/o-Fr-1| 67.01  | 89.10   | 64.67  |
> |          | w/o-Phase | 82.13  | 82.14   | 65.42  |
> | F1 (↑)   | Base   | 86.84  | 87.82   | 72.41  |
> |          | w/o-Fr-1 | 66.49  | 90.76   | 62.75  |
> |          | w/o-Phase | 81.86  | 88.91   | 63.77  |
> | ROC (↑)  | Base   | 90.08  | 91.83   | 71.44  |
> |          | w/o-Fr-1 | 72.75  | 91.42   | 71.21  |
> |          | w/o-Phase | 85.22  | 93.02   | 70.18  |
>
> A new ablation study was conducted to test whether the APCE module’s gain comes from genuine phase–amplitude interaction, rather than simple feature concatenation. We randomly shuffled phase and amplitude vectors independently for each sample, thereby destroying any correspondence. The resulting performance dropped markedly. Only the original APCE (with true cross-encoding) preserved high accuracy. This confirms that meaningful phase–amplitude interaction—not mere concatenation—is critical for the predictive power of our approach.
>
> | Dataset   | APCE (Original) | APCE (Shuffled) |
> |-----------|-----------------|-----------------|
> | Beirut  | 84.34           | 76.47           |
> | IESS   |     72.41        | 65.66           |
> | KAIME   | 86.84            | 78.57           |
> | RESPECT| 87.82            | 77.48           |
>
> **(4)** We appreciate the reviewer's suggestion of a  direct comparison with standard PAC estimators.  APCE is not a PAC estimator but an amplitude–phase fusion block inspired by PAC concepts. Nevertheless, we implemented PAC-based baselines using MVL-MI and KL-MI  computed for frequency bands of delta, theta, alpha, beta coupling with gamma (such combinations are experimentally attested PAC in epileptic brains) across channels as features processed by a two-layer MLP classifier. The results show that FAPEX significantly outperforms these baselines  (Wilcoxon rank-sum test: p<0.001):
> | Dataset | FAPEX (F1) | MVL-MI-MLP (F1) | KL-MI-MLP (F1) |
> |---------|------------|-----------------|----------------|
> | RESPECT | 87.82      | 75.46           | 68.26          |
> | KAIME   | 86.84      | 74.83           | 62.81          |
> | IESS    | 72.41      | 62.64           | 63.65          |
> | Beirut  | 84.34      | 66.69           | 69.78          |
>
> These suggest that our model achieves better performance than PAC estimator-based
> seizure prediction. Using standard PAC metrics alone cannot achieve satisfactory classification
> of preictal vs interictal samples.
>
> ### (b) **Linear Attention**
> We respectfully clarify that our **linear attention in SCA is bidirectional**,  meaning it fully accounts for interactions among all electrode channels,  developed in CV and NLP as a suboptimal yet efficient alternative.  **It does not lose the ability to capture cross-electrode dependencies**, Replacing the linear attention with full one resulted in marginal differences:
> | Dataset | Linear (F1) | Full (F1) |
> |---------|-------------|-----------|
> | Beirut  | 84.34       | 84.40     |
> | KAIME   | 86.84       | 86.87     |
> | RESPECT | 87.82       | 87.85     |
> | IESS    | 72.41       | 72.40     |
> This confirms that **linear attention is not a bottleneck for capturing inter-electrode interactions**.
>
> ### **(W2): Spatial Analysis**
> Thank you for the helpful suggestion. Attention weight analysis from the SCA module shows consistent patterns across preictal samples, reliably highlighting electrodes near known epileptogenic zones. In the FMCE dataset, for instance, patients with temporal epilepsy show strong coupling within the temporal lobe and hippocampus, with weak frontal–temporal interactions—matching SOZ annotations. These results confirm the SCA module’s effectiveness in modeling inter-electrode dependencies and will be added to the revised manuscript.
>
> We also highlight that Supplementary Section B2 (Brain Region Analysis)  already provides quantitative evidence of spatial relevance. As shown in Figure 3, masking channels from clinically critical regions (e.g., Temporal Lobe, Hippocampus) causes significant performance drops,  confirming that FAPEX effectively leverages physiologically meaningful regions rather than relying on arbitrary patterns.
>
> ### **(W3 & Q3): Efficiency Analysis**
> Thank you for this important question. We address it in two points.
> ### (a) **Efficiency**
> We present a comparison including parameter count, latency (ms) and FLOPs per inference for best-performing
> models. On average FAPEX has moderate to medium computational efficiency with higher overall performance across datasets.
>
> | Model            | Params | Latency (ms) | FLOPs   |
> |------------------|--------|--------------|---------|
> | Brant            | 68.49M | 18.39        | 37.36G  |
> | CBrMod           | 4.03M  | 8.45         | 1.29G   |
> | EEGPT            | 25.24M | 13.93        | 9.79G   |
> | Neuro-BERT       | 7.75M  | 12.08        | 2.42G   |
> | VQMTM            | 4.29M  | 9.85         | 1.45G   |
> | COMETS           | 3.95M  | 3.90         | 432.91M |
> | MF-CLR           | 4.21M  | 3.09         | 753.09M |
> | TS2Vec           | 3.95M  | 3.83         | 423.23M |
> | FAPEX-Small (SL) | 3.51M  | 5.29         | 566.42M |
> | FAPEX-Base (SL)  | 9.32M  | 9.51         | 1.35G   |
>
> ### (b) **Wearable Device**
> We thank the reviewers for raising the question of wearable EEG applicability. FAPEX is currently designed for clinical-grade settings (scalp EEG, iEEG, and animal models) under controlled conditions, as consumer-grade preictal datasets remain scarce and fragmented. As noted in the Supplementary Limitations, this makes rigorous wearable evaluation infeasible at present. Nonetheless, FAPEX demonstrates strong generalization across 12 diverse clinical datasets, with efficient parameter size and latency. Our interpretability analysis further shows that FAPEX relies on clinically relevant brain regions (e.g., temporal, frontal, hippocampus), supporting its potential for future adaptation to low-channel or wearable EEG. We view this as a promising direction and plan to pursue it as suitable benchmarks emerge.
>
> ### **(W4 & Q4): Formatting**
> We sincerely thank the reviewer for the careful reading
> of our manuscript and for pointing out the formatting issues. We have thoroughly proofread and corrected the entire paper and appendices.
>
> ## **Summary**
> We hope these additions could address your concerns
> and strengthen our submission. In light of these, we respectfully ask you to reconsider the score if the new material satisfactorily resolves the earlier reservations.

---

> > ### Comment · Reviewer_8XFQ · 2025-08-06
> >
> > I thank the authors for their detailed and constructive rebuttal.
> >
> > The responses have successfully addressed the questions and concerns I raised in my initial review. The additional clarifications and results have strengthened the paper and resolved my previous reservations.
> >
> > Given the effective rebuttal, I have raised my score to **5: Accept**.

---

> > > ### Author Response · Authors · 2025-08-06
> > >
> > > We sincerely thank the reviewer for their constructive and comprehensive advice. We greatly appreciate your supportive recommendation. Your insightful comments were instrumental in guiding significant improvements and refinements to the paper.

---

### Note · Authors · 2025-08-12

We thank the Area Chair and reviewers for their thoughtful discussion. After rebuttal, scores were updated to 5, 5, 5, and 4, reflecting a consensus on FAPEX’s novelty, rigor, and impact in subject-agnostic seizure prediction.

For clarity:  Our rebuttal addressed specific questions without altering the paper’s claims. We drew upon analyses already in the supplement or minor extensions to confirm assumptions, consistent with NeurIPS guidelines.

### • Core contribution and novelty
The Fractional Neural Frame Operator is a learnable fractional filterbank with deformation stability that reduces spectral bias and yields informative instantaneous amplitude and phase. Together with APCE and SCA, it supports heterogeneous montages and generalizes across **12 benchmarks**, **4 species**, and **4 modalities** under nested cross-validation.

### • Amplitude–phase cross-encoding (APCE)
APCE is an efficient neural fusion block inspired by physiological coupling. A shuffled-control study that broke phase–amplitude correspondence reduced performance metrics, confirming that gains arise from genuine interaction rather than concatenation.

### • Spatial correlation aggregation (SCA)
Attention maps align with clinically annotated seizure onset zones, and region masking causes clear drops in performance. Linear attention performs on par with full attention, preserving efficiency while modeling inter-electrode dependencies.

### • Clinical metrics and efficiency
We report precision, specificity, false positives per hour, and latency. FAPEX achieves about low FPR with high specificity on representative datasets, while remaining compact and fast, outperforming larger baselines.

### • Dataset and preprocessing
We clarified epoch durations, channel handling, and separate analyses for scalp EEG and iEEG. Mixed-modality pretraining improves generalization without bias. A **1024 Hz ablation** yields only slight gains, and adding ictal windows during training produces **marginal, non-significant improvements**, supporting the **preictal-only focus** used in clinical prediction protocols.

---

Given the consensus across reviewers and authors on novelty, generalization, and clinical importance, we believe the work merits **acceptance** and respectfully request **Spotlight consideration**.  The operator and architecture are broadly relevant to **time–frequency learning** and **state-space modeling** on nonstationary biosignals beyond epilepsy.

---

### Decision · Program_Chairs · 2025-09-17

**Decision:**

Accept (spotlight)

**Comment:**

The paper introduces FAPEX, a deep learning architecture for subject-agnostic seizure prediction. Its core innovation is the Fractional Neural Frame Operator, a learnable fractional-order convolutional filterbank that adaptively captures both low- and high-frequency EEG components. FAPEX further integrates an Amplitude-Phase Cross-Encoding module with bidirectional state-space models to capture amplitude–phase interactions, and a Spatial Correlation Aggregation module with linear attention to handle heterogeneous electrode setups. Evaluated across 12 diverse datasets spanning multiple species and modalities, FAPEX consistently outperforms 32 state-of-the-art baselines, with ablations and domain transfer tests confirming its strong generalization.

Strengths:
1. Most reviewers agree that the idea of fractional neural frame operator is innovative and theoretically grounded.
2. Most reviewers find that the experiments provided are convincing and extensive.

Weakness:
Some reviewers request more explanations about the implementation details, the complexity and generalizability of the model.

During the rebuttal, the authors have addressed the following questions:
1. Providing more explanations about the implementation details and the novelty of the paper.
2. Additional experiments to demonstrate the effectiveness of APCE and SCA modules
3. Additional analysis of the complexity of the proposed model.

After the rebuttal, the reviewers unanimously agreed that the paper is worthy of acceptance and that their concerns have been adequately addressed. Therefore, I recommend accepting this submission as a spotlight paper.